# The Integrity of the HMR complex is necessary for centromeric binding and reproductive isolation in *Drosophila*

**Andrea Lukacs**[1,2], **Andreas W. Thomae**[3], **Peter Krueger**[1¤a], **Tamas Schauer**[4], **Anuroop V. Venkatasubramani**[1,2], **Natalia Y. Kochanova**[1,2¤b], **Wasim Aftab**[2,5], **Rupam Choudhury**[1], **Ignasi Forne**[5], **Axel Imhof**[1,5]*

1 Biomedical Center, Chromatin Proteomics Group, Department of Molecular Biology, Faculty of Medicine, Ludwig-Maximilians-Universität München, Planegg-Martinsried, Germany, 2 Graduate School of Quantitative Biosciences (QBM), Ludwig-Maximilians-Universität München, Munich, Germany, 3 Biomedical Center, Core Facility Bioimaging, Faculty of Medicine, Ludwig-Maximilians-Universität München, Planegg-Martinsried, Germany, 4 Biomedical Center, Bioinformatics Unit, Faculty of Medicine, Ludwig-Maximilians-Universität München, Planegg-Martinsried, Germany, 5 Biomedical Center, Protein Analysis Unit, Faculty of Medicine, Ludwig-Maximilians-Universität München, Planegg-Martinsried, Germany

¤a Current address: TUM School of Medicine, Epigenetics of skin ageing, TU Munich, Garching, Germany
¤b Current address: Wellcome Centre for Cell Biology, University of Edinburgh, ICB, Michael Swann Building, King's Buildings, Max Born Crescent, Edinburgh, United Kingdom
* Imhof@lmu.de

**Data Availability Statement:** ChIP-seq were deposited to GEO with accession number GSE163058. The mass spectrometry data have

## Abstract

Postzygotic isolation by genomic conflict is a major cause for the formation of species. Despite its importance, the molecular mechanisms that result in the lethality of interspecies hybrids are still largely unclear. The genus *Drosophila*, which contains over 1600 different species, is one of the best characterized model systems to study these questions. We showed in the past that the expression levels of the two hybrid incompatibility factors *Hmr* and *Lhr* diverged in the two closely related *Drosophila* species, *D. melanogaster* and *D. simulans*, resulting in an increased level of both proteins in interspecies hybrids. The overexpression of the two proteins also leads to mitotic defects, a misregulation in the expression of transposable elements and decreased fertility in pure species. In this work, we describe a distinct six subunit protein complex containing HMR and LHR and analyse the effect of *Hmr* mutations on complex integrity and function. Our experiments suggest that HMR needs to bring together components of centromeric and pericentromeric chromatin to fulfil its physiological function and to cause hybrid male lethality.

## Author summary

A major cause of biological speciation is the sterility and/or lethality of hybrids. This hybrid lethality is thought to be the consequence of two incompatible genomes of the two different species. We used the fruit fly Drosophila melanogaster as a model system to isolate a defined protein complex, which mediates this hybrid lethality. Our data suggest that this complex containing six subunits has evolved in one Drosophila species (*Drosophila*

been deposited to the ProteomeXchange Consortium via the PRIDE partner repository with the dataset identifiers PXD023188 and PXD023193. Interactive Network and volcano plots from HMR complex components purifications are entirely available at the following https://wasim-aftab.shinyapps.io/SccNet-AL/.

**Funding:** This work was supported by grants from the Deutsche Forschungsgemeinschaft (DFG) to AI (IM23/14-1, CRC1309-TPB03, CRC1064-TPZ03) and predoctoral grants to AL, NYK, AVV and WA (QBM). The funders had no role in study design, data collection and analysis, decision to publish, or preparation of the manuscript. RC and AVV received a salary from the DFG (CRC1309-TPB03) NYK received a salary from the DFG (IM23/14-1).

**Competing interests:** The authors have declared that no competing interests exist.

*melanogaster*) to bring together components of centromeric and pericentromeric chromatin. We show that the integrity of the complex is necessary for its genomic binding patterns and its ability to maintain fertility in female *Drosophila melanogaster* flies. Hybrid males between *Drosophila melanogaster* and the very closely related species *Drosophila simulans* die because they contain elevated levels of this complex. These high levels result in mitotic defects and a misregulation in the expression of transposable elements in those hybrids. Our results show that mutations that interfere with the complex's function in *Drosophila melanogaster* also fail to induce lethality in hybrids suggesting that its evolutionary acquired functions in one species induce lethality in interspecies hybrids.

## Introduction

Eukaryotic genomes are constantly challenged by the integration of viral DNA or the amplification of transposable elements. As these challenges are often detrimental to the fitness of the organism, they frequently elicit adaptive compensatory changes in the genome. As a result of this process, the genomes as well as the coevolving compensatory factors, can rapidly diverge. Such divergences can result in severe incompatibilities eventually leading to the formation of two separate species [1,2].

Arguably the best characterized system for studying the genetics of reproductive isolation and hybrid incompatibilities is constituted by the two closely related *Drosophila* species *D. melanogaster* and *D. simulans* (*D. mel* and *D. sim*) [3]. One century of genetic studies has led to the identification of three fast evolving genes that are critical for hybrid incompatibility: *Hmr* (Hybrid male rescue), *Lhr* (Lethal hybrid rescue) and *gfzf* (GST-containing FLYWCH zinc finger protein) [4–8]. The genetic interaction of these three genes results in the lethality of *D.mel/D.sim* hybrid males. Strikingly, all three genes are fast evolving and code for chromatin proteins suggesting that their fast evolution reflects adaptations to genomic alterations. While the molecular interaction between HMR and LHR is well established in pure species as well as in hybrids [9,10], the molecular basis for their genetic interaction with GFZF is unclear. Interestingly, in interspecies hybrids or when HMR/LHR are overexpressed, HMR spreads to multiple novel binding sites many of which have been previously characterized to also bind GFZF [11].

In the nucleus of tissue culture cells and in imaginal discs, HMR and LHR form defined foci that are clustered around centromeres [9,12,13]. Super-resolution microscopy and chromatin immunoprecipitation revealed that HMR is often found at the border between centromeres and constitutive pericentromeric heterochromatin bound by HP1a [12,14,15]. In addition to pericentromeric regions, HMR also binds along chromosome arms colocalizing with known *gypsy*-like insulator elements [14]. Depending on the tissue investigated, HMR shows slightly different binding patterns. It binds to telomeric regions of polytene chromosomes [9,11], colocalizes with HP1a in early *Drosophila* embryos [10] and near DAPI-bright heterochromatin in larval brain cells [13].

Flies carrying *Hmr* or *Lhr* loss of function alleles show an upregulation of transposable elements (TEs), defects in mitosis, and a reduction of female fertility in *D. mel*. Expression of transposable elements is increased particularly in ovarian tissue but also in cultured cells [9,10]. The mechanism that causes such a massive and widespread upregulation is not entirely clear as most of the TEs that respond to a reduced *Hmr* dosage are not bound by HMR under native conditions [14]. Due to the overexpression of the *HeT-A*, *TART* and *TAHRE* retrotransposons, *Hmr* mutants show a substantial increase in telomere length [10] and an increased number of anaphase bridges during mitosis presumably due to a failure of sister chromatid

detachment during anaphase [13]. The massive upregulation of transposable elements in ovaries is possibly also the cause of the substantially reduced fertility of *Hmr* and *Lhr* mutant female flies [16].

Many of the phenotypes observed in cell lines lacking *Hmr* and *Lhr*, are mirrored by *Hmr* and *Lhr* overexpression, highlighting the importance of properly balanced *Hmr/Lhr* levels [9]. Hybrids show enhanced levels of both proteins relative to the pure species and consistently, are also characterized by loss of transposable elements silencing and cell cycle progression [9,10,17]. The latter is thought to be the cause for the failure of male hybrids to develop into adults, given the almost complete absence of imaginal discs [13,18–20]. In addition, hybrids and *Hmr/Lhr* overexpressing cells, display a widespread mis-localization of HMR at several euchromatic loci at chromosome arms including the previously unbound GFZF binding sites [9,11,12].

To better understand the deleterious effects observed in the presence of an excess of HMR, we decided to investigate the binding partners of HMR under native conditions and upon overexpression. Our results suggest that HMR belongs to a defined protein complex composed of 6 subunits under native conditions and gains novel chromatin associated interactors when overexpressed. Moreover, we show that *Hmr* mutations that interfere with complex formation lead to the loss of HMR's function in pure species and in hybrids.

## Materials and methods

### Cell culture and induction

*Drosophila melanogaster* Schneider cell lines (SL2) were grown in Schneider's medium (Gibco Life Technologies) supplemented with 10% fetal calf serum and antibiotics (penicillin 100 units/mL and streptomycin 100 μg/mL) at 26°C. Stable SL2 cells transfected with metallothionein promoter (pMT) driven *FLAG-HA-Hmr+/Myc-Lhr*, *FLAG-HA-Hmr²/Myc-Lhr*, *FLAG-HA-HmrᵈC/Myc-Lhr*, *FLAG-HA-Boh1*, *FLAG-HA-Boh2* were generated as described in (Thomae *et al*, 2013). Cells carrying inducible transgenes were selected with 20 μg/mL Hygromycin B and induced for 18–24 h with CuSO₄ before experiments. Cell lines carrying different *FLAG-HA-Hmr* transgenic alleles + *Myc-Lhr* as well as cells expressing *FLAG-HA-BOH1* were induced with 250 μM CuSO₄. For *FLAG-HA-BOH2*, 500 μM CuSO₄ was used. Wild type *FLAG-Hmr* SL2 cells were generated in [14] by CRISPR-Cas9 mediated gene editing of the endogenous *Hmr* gene.

### Recombinant protein expression

*Spodoptera frugiperda* 21 (SF21) cells (Gibco Life Technologies) were used for baculovirus-driven recombinant co-expression of *HA-Hmr* and *His-Lhr*. Nuclei were extracted essentially as in [21] and HMR/LHR co-immunoprecipitation was performed using mouse anti-HA Agarose beads (1 μL packed beads/mL SF21 culture, Sigma-Aldrich A2095).

### Cloning

Cloning of *Hmr* and *Lhr* ORFs into the pMT-FLAG-HA plasmid was described in [9]. Restriction fragments containing *Hmr* and *Lhr* ORFs were sub-cloned into pFast Bac Dual (containing an N-terminal HA-tag) and pFast Bac HTb, respectively. The resulting plasmids were transformed into DH10Bac and recombined bacmids were isolated from clonal transformants. PCR verified recombinant bacmids were used for transfection of SF21 cells. *Boh1* and *Boh2* were PCR amplified from genomic DNA, cloned into pJet1.2 and verified by sequencing.

ORFs were then sub-cloned into the pMT-FLAG-HA expression vector described in [9]. Full cloning details, plasmids sequences and plasmids are available on request.

### Nuclear extraction for immunoprecipitation

Cells were harvested, centrifuged at 1200 x g and washed with cold PBS. Cell pellets were resuspended in hypotonic buffer (10 mM Hepes pH 7.6, 15 mM NaCl, 2 mM $MgCl_2$, 0.1 mM EDTA, cocktail of protease inhibitors + 0.25 μg/mL MG132, 0.2 mM PMSF, 1 mM DTT) and incubated on ice for 20 min. Cells were incubated for another 5 min after addition of NP40 to a final concentration of 0.1% and then dounced with 20 strokes. 10% Hypertonic buffer (50 mM Hepes pH 7.6, 1 M NaCl, 30 mM $MgCl_2$, 0.1 mM EDTA) was added to rescue isotonic conditions. Nuclei were centrifuged for 10 minutes at 1500 x g and the supernatant was discarded. Nuclei were washed once in isotonic buffer (25 mM Hepes pH 7.6, 150 mM NaCl, 12.5 mM $MgCl_2$, 1 mM EGTA, 10% glycerol, cocktail of protease inhibitors + 0.25 μg/mL MG132, 0.2 mM PMSF, 1 mM DTT). After resuspension in the same buffer, nuclei were treated with benzonase (MERCK 1.01654.0001) and incubated for 30 minutes at 4°C on a rotating wheel. Soluble proteins were extracted by increasing the NaCl to 450 mM and incubated for 1 h at 4°C on a rotating wheel. Finally, the soluble material was separated from the insoluble chromatin pellet material by centrifugation for 30 min at 20000 x g and used for immunoprecipitations.

### Immunoprecipitation

Anti-FLAG immunoprecipitation was performed using 20 μL of packed agarose-conjugated mouse anti-FLAG antibody (M2 Affinity gel, A2220 Sigma-Aldrich) and were targeted either against the exogenously expressed transgenes (HMR[wt-tg], HMR[dC-tg], HMR[2-tg]) or an endogenously FLAG-tagged HMR (HMR). The other IPs were performed by coupling the specific antibodies to 30 μL of Protein A/G Sepharose beads. Each bait was targeted with at least one antibody (rat anti-LHR 12F4, mouse anti-HP1a C1A9, rabbit anti-NLP, anti-FLAG-M2 for FLAG-BOH1 and FLAG-BOH2), while HMR was targeted with three different antibodies (rat anti-HMR 2C10 and 12F1, anti-FLAG-M2 for FLAG-HMR). Rabbit anti-NLP and mouse anti-HP1a were directly incubated with the beads, while rat anti-HMR and anti-LHR were incubated with beads that were pre-coupled with 12 μL of a rabbit anti-rat bridging antibody (Dianova, 312-005-046). FLAG-IPs in non-FLAG containing nuclear extracts were used as mock controls for FLAG-IPs. For all other IPs, unspecific IgG coupled to Protein A/G Sepharose or Protein A/G Sepharose alone were used as mock controls.

The steps that follow were the same for all the immunoprecipitations and were all performed at 4°C. Antibody coupled beads were washed three times with IP buffer (25mM Hepes pH 7.6, 150 mM NaCl, 12.5 mM $MgCl_2$, 10% Glycerol, 0.5 mM EGTA) prior to immunoprecipitation. Thawed nuclear extracts were centrifuged for 10 minutes at 20000 x g to remove precipitates and subsequently incubated with antibody-coupled beads in a total volume of 500–600 μL IP buffer complemented with a cocktail of protease inhibitors plus 0.25 μg/mL MG132, 0.2 mM PMSF, 1 mM DTT and end-over-end rotated for 2 h (anti-FLAG) or 4 h (other IPs) at 4°C. After incubation, the beads were centrifuged at 400 x g and washed 3 times in IP buffer complemented with inhibitors and 3 times with 50 mM $NH_4HCO_3$ before on beads digestion.

### Sample preparation for mass spectrometry

The pulled-down material was released from the beads by digesting for 30 minutes on a shaker (1400 rpm) at 25°C with trypsin at a concentration of 10 ng/μL in 100 μL of digestion buffer

(1M Urea, 50 mM $NH_4HCO_3$). After centrifugation the peptide-containing supernatant was transferred to a new tube and two additional washes of the beads were performed with 50 μL of 50 mM $NH_4HCO_3$ to improve recovery. 100 mM DTT was added to the solution to reduce disulphide bonds and the samples were further digested overnight at 25˚C while shaking at 500 rpm. The free sulfhydryl groups were then alkylated by adding iodoacetamide (12 mg/mL) and incubating 30 minutes in the dark at 25˚C. Finally, the light- shield was removed and the samples were treated with 100 mM DTT and incubated for 10 minutes at 25˚C. The digested peptide solution was then brought to a pH~2 by adding 4 μL of trifluoroacetic acid (TFA) and stored at -20˚C until desalting. Desalting was done by binding to C18 stage tips and eluting with elution solution (30% methanol, 40% acetonitrile, 0.1% formic acid). The peptide mixtures were dried and resuspended in 20 μL of formic acid 0.1% before injection.

## Sample analysis by mass spectrometry

Peptide mixtures (5 μL) were subjected to nanoRP-LC-MS/MS analysis on an Ultimate 3000 nano chromatography system coupled to a QExactive HF mass spectrometer (both Thermo Fisher Scientific). The samples were directly injected in 0.1% formic acid into the separating column (150 x 0.075 mm, in house packed with ReprosilAQ-C18, Dr. Maisch GmbH, 2.4 μm) at a flow rate of 300 nL/min. The peptides were separated by a linear gradient from 3% ACN to 40% ACN in 50 min. The outlet of the column served as electrospray ionization emitter to transfer the peptide ions directly into the mass spectrometer. The QExactive HF was operated in a Top10 duty cycle, detecting intact peptide ion in positive ion mode in the initial survey scan at 60,000 resolution and selecting up to 10 precursors per cycle for individual fragmentation analysis. Therefore, precursor ions with charge state between 2 and 5 were isolated in a 2 Da window and subjected to higher-energy collisional fragmentation in the HCD-Trap. After MS/MS acquisition precursors were excluded from MS/MS analysis for 20 seconds to reduce data redundancy. Siloxane signals were used for internal calibration of mass spectra.

## Proteomics data analysis

For protein identification, the raw data were analyzed with the Andromeda algorithm of the MaxQuant package (v1.6.7.0) against the Flybase reference database (dmel-all-translation-r6.12.fasta) including reverse sequences and contaminants. Default settings were used except for: Variable modifications = Oxidation (M); Unique and razor, Min. peptides = 1; Match between windows = 0.8 min. Downstream analysis on the output proteinGroups.txt file were performed in R (v4.0.1). If not otherwise stated, plots were generated with ggplot2 package (v3.3.2). Data were filtered for Reverse, Potential.contaminant and Only.identified.by.site and iBAQ values were $log_2$ transformed and imputed using the R package DEP (v1.10.0, impute function with following settings: fun = "man", shift = 1.8, scale = 0.3). Except for Figs where data were bait normalized, median normalization was performed. Statistical tests were performed by fitting a linear model and applying empirical Bayes moderation using the limma package (v3.44.3). AP-MS for HMR complex identification (Figs 1 and S1) were compared with a pool of all control samples (IgG and FLAG mock IPs). For Figs 1C and S1E enriched proteins from AP-MS experiments from HMR complex components were first selected (cut off: log2FC > 2.5, p-adjusted < 0.05) and then intersection was quantified and plotted with UpsetR (v1.4.0). The Network graph in S1E Fig was prepared with force directed layout in D3.js and R. The network graph was prepared using Cytoscape (3.4.0) with input from all AP-MS experiments and the String database (vs11.0) [22]

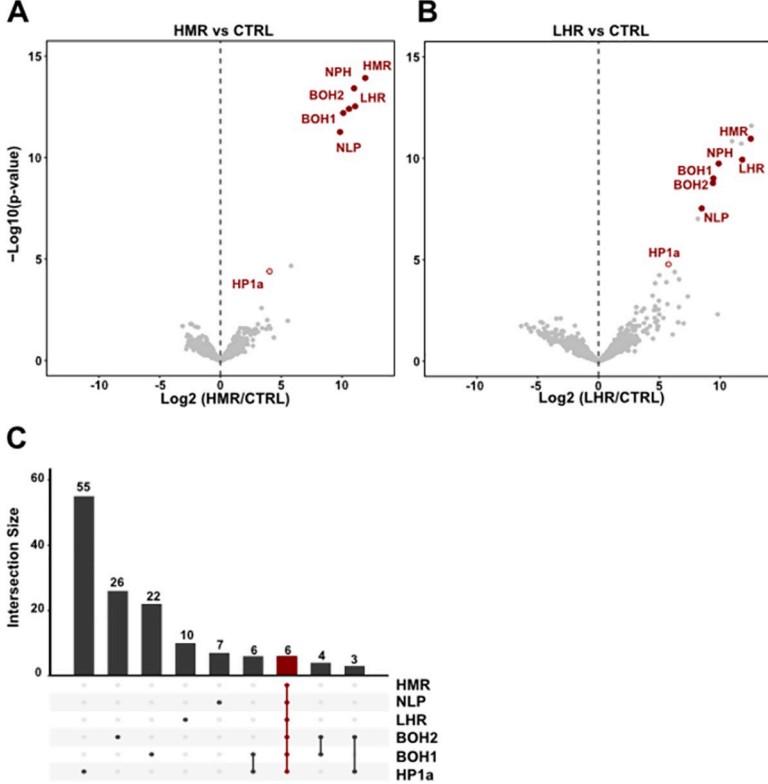

**Fig 1. The hybrid incompatibility proteins HMR and LHR interact in a defined HMR/LHR complex.** (**A**) and (**B**) Volcano plots highlighting reproducible interactors of HMR and LHR. Interactors consistently found in both HMR and LHR AP-MS are labelled in red. X-axis: $\log_2$ fold-change of factor enrichment in HMR (left, n = 8) or LHR (right, n = 4) IPs against mock purification (CTRL). Y-axis: significance of enrichment given as $-\log_{10}$ p-value calculated with a linear model. (**C**) The 6 HMR complex subunits are the only set of proteins shared in all HMR, LHR, NLP (n = 3), BOH1 (n = 4), BOH2 (n = 5) and HP1a (n = 4) AP-MS experiments (in red). Enriched proteins from each AP-MS experiment from the different complex components were first selected (cut off: log2FC > 2.5, p-adjusted < 0.05). Subsequently, the intersection among such enriched interactors was calculated and plotted with the UpsetR package. The intersection plot shows the number of interactors (bars) that are unique to one or more of the subsets representing the interactomes resulting from different IPs (rows). Lines connected dots define specific intersections between two or more interactomes. Unlabelled additional bait-specific interactors from (**A**) and (**B**) are available in S2 Table or in interactive plots at (URL). Additional volcano plots from the HMR complex components AP-MS are shown in S1 Fig.

## Western blot analysis

Samples were boiled 10 min at 96˚C in Laemmli sample buffer, separated on SDS-PAGE gels, processed for western blot using standard protocols and detected using rat anti-HMR 2C10 (1:20), rat anti-LHR 12F4 (1:20), mouse anti-HP1a C1A9 (1:20), mouse anti-HA 12CA5 (1:20), mouse anti-FLAG SIGMA-M2 (1:1000), mouse anti-Lamin (1:1000) antibodies. Secondary antibodies included sheep anti-mouse (1:5000) (RRID: AB772210), goat anti-rat (1:5000) (RRID: AB772207), donkey anti-rabbit (1:5000) (RRID: AB772206) coupled to horseradish peroxidase. For proteins detection after IP, beads were boiled in Laemmli sample buffer after washing. For protein detection in ovaries, a short nuclear extraction was performed from 10 pairs of ovaries prior to boiling in sample buffer.

## ChIP-Seq

Chromatin immunoprecipitation was essentially performed as in [14]. For each anti-FLAG ChIP reaction, chromatin isolated from $1–2 \times 10^6$ cells were incubated with 5 μg of mouse

anti-FLAG (F1804, SIGMA-ALDRICH—RRID: AB262044) antibody pre-coupled to Protein A/G Sepharose. For ChIPs targeting total HMR, the same amount of chromatin was incubated with rat anti-HMR 2C10 antibody pre-coupled to Protein A/G Sepharose through a rabbit IgG anti-rat (Dianova, 312-005-046). Samples were sequenced (single-end, 50 bp) with the Illumina HiSeq2000. Sequencing reads were mapped to the *Drosophila* genome (version dm6) using bowtie2 (version 2.2.9) and filtered by mapping quality (-q 2) using samtools (version 1.3.1). Sequencing depth and input normalized coverages were generated by Homer (version 4.9). Enriched peaks were identified by Homer with the parameters -style factor -F 2 -size 200 for each replicate.

High confidence FLAG-HMR peaks (a pool of HMR^wt-tg and HMR^dC-tg) were called when a peak was present in at least half of the samples (5 out of 10). Coverages were centred at high confidence FLAG-HMR peaks in 4 kb windows and binned in 10 bp windows. The as-such generated matrices were z-score normalized by the global mean and standard deviation. HP1a-proximal peaks were defined as 10 percent of the peaks with highest average HP1a ChIP signal in 4 kb windows surrounding peaks. Composite plots and heatmaps indicate the average ChIP signal (z-score) across replicates. Heatmaps were grouped by HP1a class and sorted by the average ChIP signal in HMR native in a 400 bp central window. For statistical analysis, the average ChIP signal (z-score) was calculated in a 200 bp central window across peaks for each replicate. P-values were obtained by a linear mixed effect model (R packages: lme4 version 1.1–23 and lmerTest version3.1–2), in which average ChIP signal was included as outcome, genotype (*Hmr*^+ or *Hmr*^dC) and peak class (HP1a-proximal or non-HP1a-proximal) as fixed effects and sample ids as random intercept.

Chromosome-wide coverage plots were generated by averaging replicates, binning coverages in 50 kb windows and z-score normalizing by the global mean and standard deviation.

The percentage of peaks on chromosome 4 relative to the total number of peaks was calculated for each replicate. P-value was obtained by a linear model (R package: stats version 3.6.1), in which percentage was included as outcome and genotype (*Hmr*^+ or *Hmr*^dC) as independent variable.

## Immunofluorescent staining in SL2 cells

Immunofluorescent staining of SL2 cells was performed as described previously [9,12]. For Fig 4A the following antibodies were used: mouse anti-HA 12CA5 1:1000 (Roche), rat anti-HMR 2C10 1:25 (Helmholtz Zentrum Munich), rabbit anti-dCENP-C 1:5000 (kind gift from C. Lehner) as a centromeric marker. For S2A Fig: mouse anti-HA (Invitrogen 2–2.2.14; 1:300), rabbit anti-CENP-A (Active Motif, 1:300), rat anti-HMR (2C10; 1:25); For S2B Fig: rat anti-HA (Sigma Aldrich 3F10; 1:100), mouse anti-HP1a (DSHB C1A9, 1:100), rabbit anti-CENP-A (1:300). Figs 4A and S6: confocal microscopy z-scans were done on a Leica TCS SP5 (with 63x objective with 1.3 NA) with a step of 0.25 μM. Sum intensity projections were analyzed, and only cells with a minimum nucleoplasmic intensity of 70 a.u. on the anti-HMR channel were taken into account for further analysis. Two different quantifications were performed (respectively Figs 4A and S6). In one case cells were separated and counted based on the degree of co-localization between HMR and CENP-C: overlapping, partially overlapping or non-overlapping. In parallel, the number of CENP-C marked centromeric foci associated with HMR signal was measured. Both cells and centromeric foci were blind-counted, the experiment was repeated in 2 biological replicates and for each replicate at least two slides were measured (for each slide between 24 and 63 cells were quantified). Further details about stainings for S2 Fig and microscopy are available as S1 Methods.

## Immunofluorescent staining in ovaries

Flies were grown at 25°C for 7–9 days and fed in yeast paste for at least 3 days prior to dissection. Ovaries were dissected in ice-cold PBS, then ovarioles were teased apart with forceps and moved to 1.5 mL tubes. PBS was removed and fixation solution (400 μL PBS Paraformaldehyde 2%, Triton 0.5%, 600 μL Heptane) was added. Samples were incubated for 15 min at room temperature on a rotating wheel. Fixed ovaries were washed 3 times with 200 μL of PBS-T and then blocked for 30 min at room temperature on a rotating wheel (PBS-T, NDS 2%). After rinsing, 100 μL primary antibody solution (PBS-T, rat anti-HMR-2C10 1:20 mouse anti-HP1a-C1A9 1:10 and NDS 2%) were added and samples were incubated rotating overnight at 6°C. Primary antibody was washed three times with PBS-T. Secondary antibody solution was added (200 μL PBS-T + donkey anti-mouse Alexa 488 1:600, donkey anti-rat Cy3 1:300 and 2% NDS) and samples were rotated for 2 h at room temperature. Samples were washed three times with PBS-T and incubated for 10 min at room temperature with 200 μL of DAPI 0.002 mg/mL. Following this, they were washed once with PBS-T and once with PBS. Stained ovaries were finally mounted with one drop of vectashield in epoxy diagnostic slides (Thermo-Fisher Scientific, 3 wells 14 mm) and covered with high precision cover glasses. Further details about stainings for Figs 5 and S7 and microscopy are available as S1 Methods.

## *Drosophila* husbandry and stocks

*Drosophila* stocks were reared on standard yeast glucose medium and raised at 25°C on a 12h/12h day/night cycle. For the transgenic fly lines the entire *D.mel* genomic region including the *melanogaster-Hmr* gene and parts of the flanking CG2124 and Rab9D genes (a 9538 bp fragment: X10,481,572–10,491,109), was cloned into a plasmid backbone containing a mini-white gene and a p-attB site. Plasmids for control $Hmr^+$ stocks contained a wild type copy of *Hmr* while plasmids for test stocks contained either of the mutated versions $Hmr^{dC}$ or $Hmr^2$. $Hmr^{dC}$ plasmids carry a point mutation with an A-T substitution (base 3667 of the CDS) that turns a Gly into a premature STOP codon and results in a C-terminally truncated protein product (last 171 aa missing). $Hmr^2$ plasmids carry the two point mutations E371K and G527A (described in [16,23]). Identity of constructs was confirmed by sequencing. PhiC31 integrase-mediated transformations of the *D. melanogaster* line y1 w67c23; P{CaryP}attP2 (BL8622) were performed by BestGene Inc. resulting in the transgenic integration in attP2 docking site in chromosome 3 (3L:11070538). All rescue experiments were performed by crossing transgenes into *Df(1)Hmr-*, *ywv* background [16]

## Crosses for generating *Hmr* genotypes for complementation tests in *D. melanogaster*

Males y w; *Hmr\*/TM6, Tb* (*Hmr\** = any *Hmr* transgenic allele, including $Hmr^+$, $Hmr^{dC}$, $Hmr^2$) were crossed to *Df(1)Hmr-* females. F1 males *Df(1)Hmr-/Y; Hmr\*/+* were backcrossed to the same females *Df(1)Hmr-* to generate females *Df(1)Hmr-/Df(1)Hmr-; Hmr\*/+*. The latter females were crossed with their sibling males *Df(1)Hmr-/Y; Hmr\*/+* to generate homozygotes for *Hmr\** used for ovaries immunofluorescent HMR/HP1a stainings. To obtain isogenic flies for fertility and retrotransposon silencing complementation assays, heterozygous females *Df(1)Hmr-; Hmr\*/+* were outcrossed to males *Df(1)Hmr-/Y* for 3–8 or 7–8 generations, respectively. Isogenized flies *Df(1)Hmr-; Hmr\*/+* were crossed to *Df(1)Hmr-* and the resulting siblings were compared: control non-complemented individuals *Df(1)Hmr-* and test individuals *Df(1)Hmr-; Hmr\*/+*.

## Crosses for generating *Hmr* genotypes for hybrid viability assays

Young *D.mel* females *Df(1)Hmr-*; *Hmr** were crossed at 25˚C to 1–5 day old wild type *D.sim* males (C167.4 or $w^{501}$). Control *D.mel* stocks *Df(1)Hmr-* were crossed in parallel to *D.sim* males (control cross: no lethality rescue). Crosses were transferred regularly to fresh medium. When larval tracks became visible, vials were transferred at 20˚C to improve recovery of interspecific hybrids. Vials were kept until the last adults eclosed and number and genotype of hybrid offspring was scored. Rescue was measured by counting the number of viable transgene carrying males from the corresponding cross. In the rescue experiment, *Hmr*+ served as a positive (lethality rescue) and *Hmr*$^2$ as a negative control (no lethality rescue). For statistical testing, Wilcoxon rank sum test (non-parametric) was used for pairwise comparisons with FDR correction for multiple testing using ggpubr package *(v0.4.0*, using compare_means function with following settings: formula = percent_males_offspring ~ Hmr_allele, method = 'wilcox.test', p.adjust.method = 'fdr').

## Fertility assays

Three 1–3 days old *D. melanogaster* females were crossed for 2–3 days with six wild type males *D. melanogaster*. Flies were then transferred to fresh vials and again every 5 days for 3 times in total. Vials were scored 15–18 days after first eggs were laid, to make sure all adults were eclosed but no F2 was included. Vials in which one female or more than one male was missing were not scored. The whole assay was performed at 25˚C. Tested females *Df(1)Hmr-*; *Hmr**/+ (*Hmr** = an *Hmr* transgenic allele) were always grown with and compared to their respective control siblings *Df(1)Hmr* -;+/+, obtained from crosses between *Df(1)Hmr-* and *Df(1)Hmr-*; Hmr*/+. Rescue was measured as total offspring counted per female. In the rescue experiment, *Hmr*+ served as a positive control (fertility rescue) and Hmr$^2$ as a negative control (no fertility rescue). For statistical testing, Wilcoxon rank sum test (non-parametric) was used for pairwise comparisons with FDR correction for multiple testing using ggpubr package *(v0.4.0*, using compare_means function with following settings: formula = offspring_per_mother ~ Hmr_allele, group.by = 'day', method = 'wilcox.test', p.adjust.method = 'fdr').

## RNA extraction, cDNA synthesis and quantitative RT-PCR

2–10 pairs of ovaries were homogenized in Trizol (Thermo Fisher; cat. no. 15596026) and processed according to the manufacturer's instructions. RNA concentration and $A_{260/280}$ ratio were measured by NanoDrop. 1 μg of RNA was treated with DNase I recombinant, RNase-free (Roche; cat no.04716728001) followed by cDNA synthesis using SuperScriptIII system (Invitrogen; cat. no: 18080051), both following the respective manufacturer's protocols. For qPCR, equal volumes of cDNA for each sample were mixed with Fast SYBR Green Master Mix (Applied Biosystems; cat. no: 4385610) and run in LightCycler 480 Instrument II (Roche; cat. no: 05015243001) in a 384-well setup. Three technical replicates were used for each sample with 18s rRNA as the housekeeping gene. Annealing temperature for all the tested primers was 60˚C and the list of primers used are given on request.

Plots for qPCR results were generated with R using ggplot2 package. For statistical testing Welch t-test was used with FDR correction for pairwise comparisons using rstatix package (v0.5.0).

## Results

### Characterization of a distinct HMR protein complex in *Drosophila melanogaster*

To identify the proteins interacting with the hybrid incompatibility (HI) proteins HMR and LHR under native conditions, we used specific monoclonal antibodies targeting HMR and

LHR to perform affinity purification coupled with mass spectrometry (AP-MS) from nuclear extracts prepared from *D. mel* SL2 cells. For HMR, we additionally validated our results by performing AP-MS with a FLAG antibody in cells carrying an endogenously tagged HMR (HMR^endo) [14]. These experiments revealed the existence of a set of four stable protein interactors shared between HMR (Fig 1A) and LHR (Fig 1B). Besides HMR and LHR, this six-subunit complex contains nucleoplasmin (NLP) and nucleophosmin (NPH) as well as two poorly characterized proteins, CG33213 and CG4788, which we named Buddy Of HMR 1 (BOH1) and Buddy Of HMR 2 (BOH2), respectively. We confirmed the complex composition by AP-MS experiments using antibodies recognizing the individual subunits, with which we immunoprecipitated all subunits of the complex (Figs 1C and S1A–S1E and S2 Table). Moreover, all subunits largely colocalize in SL2 cells, which further supports our findings (S2 Fig and [15]). HMR and LHR have also been shown to interact and colocalize with the heterochromatin protein 1a (HP1a) [9,10,12,24,25], which is consistent with our finding that both proteins as well as all other complex subunits interact with HP1a (Figs 1C, S1A–S1E and S2 Table). However, in all AP-MS experiments HP1a is present in lower amounts than the other components of the complex. This may be due to the fact that it is not a stable component or only present in a fraction of the complexes.

Similar to HP1a, the individual subunits of the complex are very likely also components of other protein complexes as we detect multiple proteins interacting exclusively with one or two components of the HMR complex (Figs 1C and S1, and S2 Table). In summary, our AP-MS results reveal the existence of a stable HMR/LHR containing protein complex under physiological conditions. As most subunits also contribute to other complexes, we wondered whether a surplus of HMR and LHR would affect complex composition.

## Overexpression of HMR and LHR results in a gain of novel protein-protein interactions

The importance of a physiological HMR and LHR dosage has been shown in non-physiological conditions like interspecies hybrids or *D. melanogaster* cells where they are artificially co-overexpressed. The increased dosage of the two proteins results in their extensive genomic mislocalization [9,11,12]. We therefore hypothesized that the overexpression of HMR and LHR in pure species also results in a gain of novel interactions, which are potentially responsible for the novel binding pattern observed. Indeed, we confirmed 30 chromatin proteins that display a stronger interaction with HMR upon HMR and LHR overexpression ([9], Figs 2A and S3 and S2 Table). These novel interactors include several proteins important for chromatin architecture such as the insulator proteins CP190, SU(HW), BEAF-32, IBF2 and HIPP1 or the mitotic chromosome condensation factor PROD. Thirteen of the novel interactors contain zinc finger DNA binding domains and three a MYB/SANT domain similar to HMR (Fig 1C and S3 Table). Intriguingly, one of the novel interactors is the product of the recently discovered missing third hybrid incompatibility gene *gfzf*. Finding GFZF as an HMR interactor under expression conditions that resemble the situation in hybrids, provides a molecular explanation for its aberrant co-localization with HMR both in hybrids and upon overexpression of HMR/LHR in tissue culture cells [11].

We also found that the ratio between HMR/LHR and the other complex components is lower in AP-MS experiments of ectopically expressed HMR/LHR (Fig 2B and 2C). Notably, under these conditions the HMR/LHR/HP1a ratio is less affected by HMR/LHR overexpression than the interactions with NLP, NPH, BOH1 and BOH2.

Establishing to which extent these newly acquired interactors or the formation of a functional HMR complex contributes to HMR/LHR's physiological function and their lethal

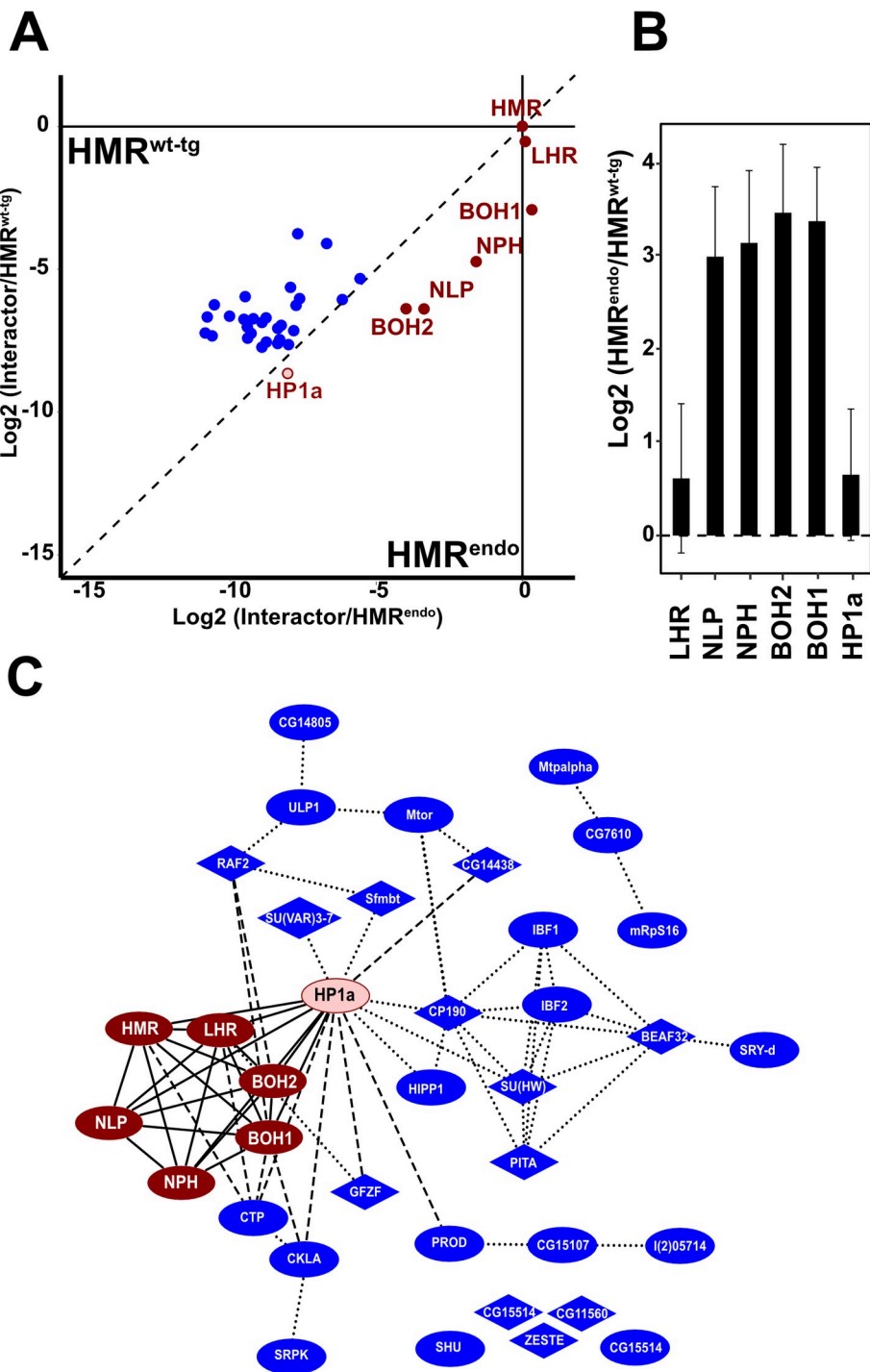

**Fig 2. Overexpression results in additional interactions of excess HMR beyond the HMR core complex. (A)** Differential interaction proteome between endogenously FLAG tagged HMR (HMR$^{endo}$, n = 4) and ectopically expressed HMR (HMR$^{wt-tg}$, n = 9). Only proteins enriched in HMR$^{wt-tg}$ or HMR$^{endo}$ vs CTRL (p < 0.05) were considered. Components of the HMR core complex and HP1a are shown in red, all other factors in blue. To display the differences within the HMR$^{endo}$ and HMR$^{wt-tg}$ interactomes, the enrichment of each putative interactor (Log2 (iBAQ$^{HMR*}$/iBAQ$^{control}$)) was normalized to the enrichment of the HMR protein used as bait. The resulting values were then plotted against each other. Dots below the diagonal indicate a stronger enrichment in the HMR$^{endo}$ pull down, the dots above the diagonal a stronger enrichment in the ectopically expressed HMR. (**B**) Differences of the ratio between HMR and members of the HMR complex and HP1a with and without ectopic expression of HMR. Plotted is the relative enrichment of each HMR complex member to the enrichment of the HMR used as bait (= the

offset from the diagonal). Error bars reflect the standard error of the means (SEM). (**C**) Network diagram of all factors enriched in the AP-MS experiments of HMR$^{endo}$ or HMR$^{wt\text{-}tg}$ (p < 0.05). Red nodes represent HMR complex components and HP1a, blue nodes additional HMR binders. Nodes containing Zn-finger domains have a diamond shape. Solid edges connect the HMR complex and HP1a, dotted edges reflect protein-protein interactions predicted using the string database [22], dashed edges reflect protein-protein interaction identified by AP-MS experiments performed in this work (Fig 1 and S3 Table). In (**A**) and (**B**) proteins were labelled only if enriched in HMR$^{wt\text{-}tg}$ or HMR$^{endo}$ vs CTRL (p < 0.05).

function in male hybrids, would provide further mechanistic details. To this end, we investigated the HMR interaction proteome upon ectopically expressing mutant HMR proteins in *D. melanogaster* SL2 cells.

## Two different *Hmr* mutations interfere with HMR complex formation and HMR localization

Most of the *Hmr* alleles that rescue hybrid male lethality are either null mutations or mutations that dramatically reduce the level of HMR (*Df(1)Hmr*⁻, *Hmr¹*, *Hmr³*, [4,9,16]) and therefore don't provide further mechanistic insights regarding Hmr's role in hybrid incompatibility. The *Hmr²* loss of function allele, however, just carries two point-mutations: one within *Hmr*'s third of four MADF domains and one in an unstructured region of *Hmr*. This third MADF domain is unusual in that it is predicted to be negatively charged and possibly mediates chromatin interactions rather than DNA binding [16]. Our previous experiments showed that HMR$^{2\text{-}tg}$ mislocalizes when expressed in SL2 cells [9]. To test whether these phenotypes can be explained by altered interaction partners, we expressed a FLAG-tagged HMR protein carrying the point mutations found in *Hmr²⁻tg* together with Myc-LHR in SL2 cells. A comparison of the interactome of ectopically expressed HMR² and HMR wildtype FLAG fusion suggests that this mutation disrupts the interaction between HMR and NLP, NPH, BOH1 or BOH2 while maintaining its interaction with LHR and HP1a (Fig 3A). Interestingly most of the factors that are picked up by ectopic HMR appear to be more represented in HMR$^{2\text{-}tg}$ than in wildtype HMR purifications (Fig 3B) suggesting that the interaction with novel interactors is not sufficient for HMR mediated lethality in hybrids.

Considering that the genetic interaction of *Hmr* and *Lhr* is critical for hybrid lethality [7], and the previously established physical interaction between the two proteins, we asked whether interfering with their physical interaction would result in a loss of function. As the C-terminal BESS domain of HMR has been suggested to be responsible for the HMR/LHR interaction [7,10,26], we recombinantly expressed either wild type HMR or C-terminally truncated HMR (HMR$^{dC\text{-}tg}$) together with LHR using a baculovirus expression system. Supporting previous evidence of a direct HMR/LHR interaction mediated by HMR BESS domain, we could co-purify LHR only with full length HMR (S4A Fig). Consistent with this observation *in vitro*, HMR$^{dC\text{-}tg}$ also shows a substantial reduction of interaction with LHR and HP1a when expressed in SL2 cells (Figs 3C and S4B and S4C) while it still interacts with NLP, NPH, BOH1 and BOH2. Besides, a reduced interaction with LHR and HP1a, HMR$^{dC\text{-}tg}$ also interacts less efficiently with other heterochromatin components that are picked up upon HMR overexpression such as SU(VAR)3-7, HIPP1, PROD or CP190 (S4D–S4G Fig and S3 Table). These findings suggest that the deletion of the BESS domain does not lead to a complete disintegration of the HMR complex but specifically interferes with its binding to LHR, HP1a and other heterochromatin proteins. Therefore, the use of the *Hmr$^{dC\text{-}tg}$* mutant allele allows us to selectively test for the functional importance of the HMR interaction with LHR and heterochromatin components.

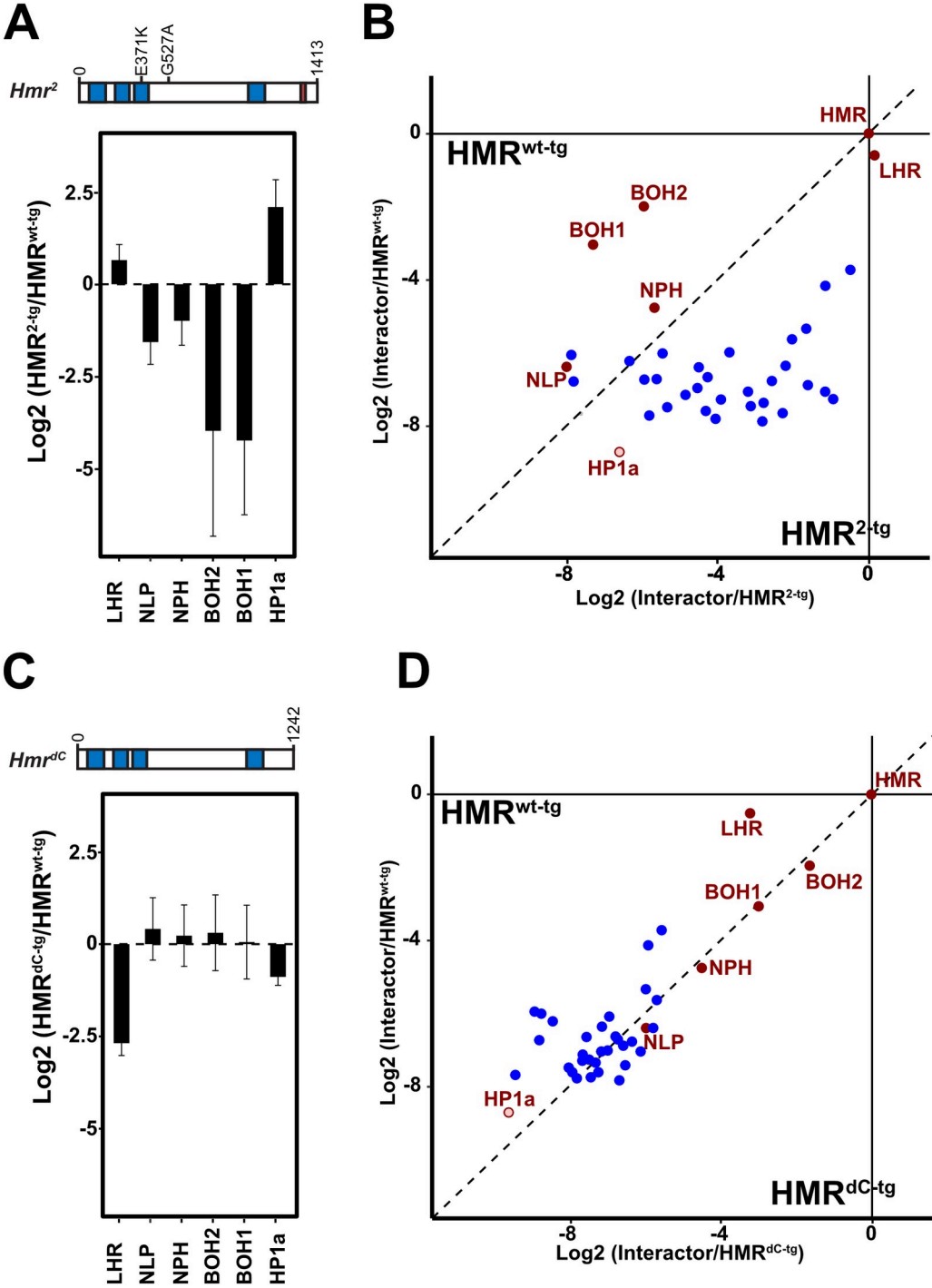

**Fig 3. Two different *Hmr* mutations interfere differently with HMR interactome and HMR complex formation.**
Effect of *Hmr²⁻ᵗᵍ* (**A**) and *Hmr^(dC-tg)* (**C**) mutations on the HMR interaction with the HMR complex components and HP1a. Y-axis represents the Log2 fold-change of HMR²⁻ᵗᵍ/HMRwt-tg and HMR^(dC-tg)/HMRwt-tg, respectively, calculated after normalization of each sample to the enrichment of the HMR protein used as bait. Error bars reflect the SEM (HMRwt-tg: n = 9; HMR^(dC-tg): n = 10, HMR²⁻ᵗᵍ: n = 3). Differential interaction proteome between ectopically expressed wild type or mutated HMR (HMRwt-tg versus HMR²⁻ᵗᵍ (**B**) or HMRwt-tg versus HMR^(dC-tg) (**D**)). Only proteins enriched in HMRwt-tg or HMRendo vs CTRL($p < 0.05$) are shown. Components of the HMR complex are shown in red, all other factors in blue. To display the differences within each interactome, the enrichment of each putative interactor was normalized to the enrichment of the HMR protein used as bait. The resulting values were then plotted against each other. Dots above the diagonal indicate a stronger enrichment in the HMRwt-tg pull down, dots below the diagonal a stronger

enrichment in the HMR mutated alleles (HMR$^{2-tg}$ or HMR$^{dC-tg}$). All proteomic data are available at the PRIDE partner repository [39] Further details are available in S3 Table.

We next wondered whether the HMR C-terminal truncation and its concomitant loss of interaction with LHR and HP1a would influence its nuclear localization. A co-staining with antibodies against the exogenously expressed HMR and a centromeric marker (anti-CENP-C) revealed a rather diffuse nuclear localization of HMR$^{dC-tg}$ in SL2 cells, which is in sharp contrast to the full length HMR, which forms distinct bright (peri)centromeric foci (Figs 4A and S6A) [9,12].

To investigate the binding of ectopic HMR$^{dC-tg}$ to the genome, we performed a genome-wide ChIP-Seq profiling of HMR$^{dC-tg}$. HMR has been previously shown to have a bimodal binding pattern in SL2 cells [14]. One class of binding sites is found in proximity to HP1a containing heterochromatic regions whereas a second class is found along chromosome arms associated with *gypsy* insulators. Consistent with the failure to interact with HP1a, HMR$^{dC-tg}$ chromatin binding is specifically impaired at HP1a-dependent sites (Figs 4B, 4C and S5), leading to a substantial reduction of HMR$^{dC-tg}$ binding in proximity to centromeres in both metacentric chromosomes 2 and 3 (Fig 4B and 4C) as well as throughout the mostly heterochromatic chromosome 4 (S5C and S5D Fig).

Altogether our results show that while the HMR C-terminus is required for HMR's interaction with LHR and HP1a and for localization in close proximity to centromeres, it is

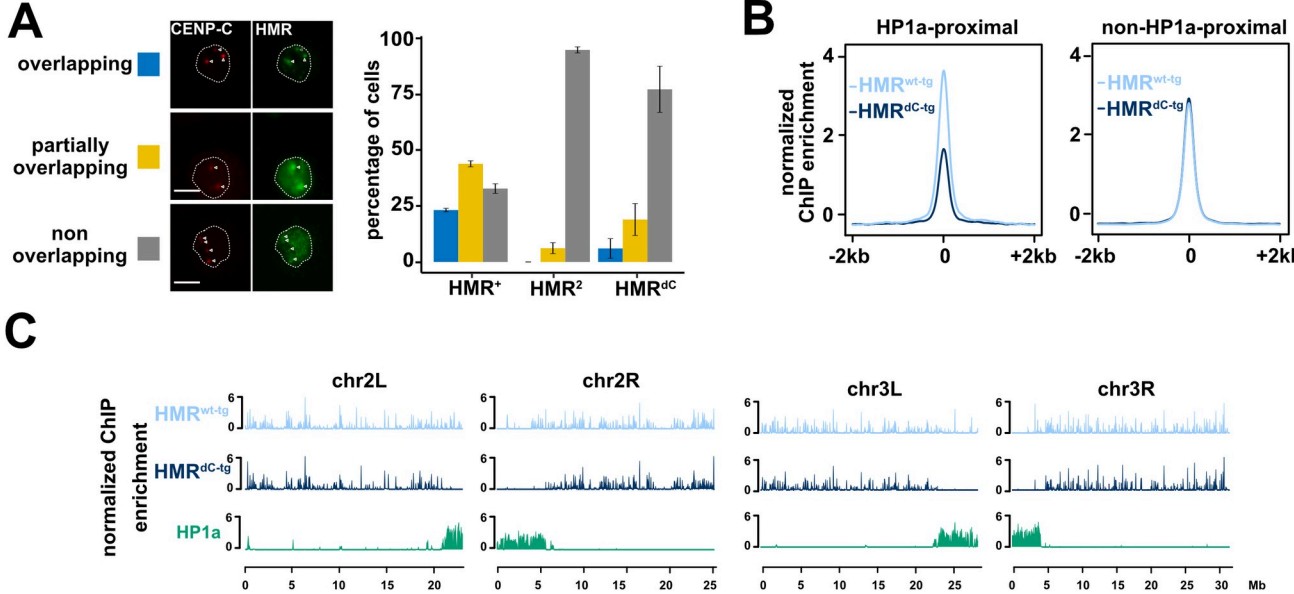

**Fig 4. The HMR C-terminus is required for HMR localization in proximity to centromeres and HP1a-bound chromatin.** (**A**) Ectopic HMR$^{dC-tg}$ fails to form bright (peri)centromeric foci in SL2 cells. Immunofluorescence images of cells expressing different *Hmr* transgenic alleles (HA-*Hmr*$^{wt-tg}$, HA-*Hmr*$^{dC-tg}$ or HA-*Hmr*$^{2-tg}$) together with wild type LHR showing the co-staining of HA-HMR and CENP-C. Based on the overlap between HMR and CENP-C signals, cells were categorized in three groups (overlapping (blue), partially overlapping (yellow) and non-overlapping (grey)) and the number of cells belonging to each group quantified. The nuclear boundary is indicated by the white dashed line and the centromeres (as identified by a-CENP-C staining) by white arrows. Error bars represent standard error of the means (n = 2). Size bars indicate 5 μm. (**B**) HP1a-proximal binding is specifically disrupted in HMR$^{dC-tg}$. Average plot of FLAG-HMR ChIP-seq profiles (z-score normalized) centred at high confidence FLAG-HMR peaks in 4 kb windows. HP1a-proximal (left) and non-HP1a-proximal (right) peaks are shown for HMR$^{wt-tg}$ (light blue) and HMR$^{dC-tg}$ (dark blue). (**C**) HMR$^{dC-tg}$ genome-wide binding is impaired in proximity to centromeres and mostly unaffected at chromosome arms. Chromosome-wide FLAG-HMR ChIP-seq profiles (z-score normalized) for HMR$^{wt-tg}$ (light blue), HMR$^{dC-tg}$ (dark blue) and HP1a (green). Chromosomes 2L, 2R, 3L and 3R are shown. Pericentromeric heterochromatin is marked by HP1a enriched territories distal (2L/3L) or proximal (2R/3R) to the respective x-axis. Plots in (**B**) and (**C**) represent an average of 5 biological replicates of FLAG-HMR ChIP-Seq in SL2 cells.

dispensable for HMR's binding to NLP, NPH, BOH1 and BOH2 and to genomic loci unrelated to HP1a. The *Hmr²* mutation in contrast does not affect HMR's ability to interact with LHR/HP1a but weakens the interaction with the other complex components. The fact that both mutations impair the centromere proximal binding suggests that complex integrity is necessary for HMR's genomic localization.

## HMR subnuclear localization changes during Drosophila oogenesis

It has been debated whether the HMR distribution we observe in SL2 cells reflects the physiological situation in flies [9,13,27]. In flies, HMR has been shown to bind to telomeric regions of polytene chromosomes [9,11], to mostly colocalize with HP1a in early *Drosophila* embryos [10] and larval brain cells [13] and to the centromere in imaginal disc cells [9]. Even in SL2 cells, where most of the HMR protein localizes close to the centromere, we detect some HMR foci that do not colocalize with centromeric foci and viceversa [12]. Unfortunately, all these somewhat contradictory results come from experiments performed under different conditions and with different antibodies, making it hard to compare the results. To have a more comprehensive view HMR's localization in flies, we used *Drosophila* ovaries as a model organ. This tissue contains different cell types at different stages of development, allowing the study of the distribution of HMR in comparable conditions within the same experiment.

Ovaries are constituted of several ovarioles containing different developmental stages that mature in an anterior to posterior direction. Both somatic and germline stem cells originate from the germarium, at the anterior tip of the ovariole (Fig 5A). Germline stem cells (GSCs) divide asymmetrically to produce another stem cell and a germline cyst cell (GCC). The GCC undergoes 4 meiotic divisions to form a cyst of 16 cells one of which will differentiate in the oocyte. The others become polyploid nurse cells that feed the oocyte. The GSCs and the differentiated cyst, are surrounded by a layer of somatic cells termed escort cells (ESCs) and follicle cells (FCs), respectively [28,29]. In most cells, HMR colocalizes with HP1a-containing pericentromeric heterochromatin (Figs 5A and S7). However, in both FCs and GSCs we also observe a colocalization of HMR with the CENP-C labelled centromeric region (Figs 5A and S7). After migration to the posterior part of the germarium, the encapsulated egg chamber matures into a single oocyte surrounded by polyploid nurse cells and follicle cells. In both stage 3 oocytes and nurse cells, which have not fully polyploidized yet, CENP-C marked centromeres are still well visible and HMR colocalizes at least partly with them (Figs 5B and 5C).

Following FCs development, we observe that as long as they are mitotically cycling (in early-stage egg chambers) HMR mostly colocalizes with CENP-C [9,10,12,13]. However, in late-stage egg chambers, post-mitotic and endoreplicating FCs show virtually no CENP-C signal and HMR localizes primarily to HP1a enriched domains (Fig 5D). All together our results show that within the same tissue, HMR can localize to centromeres in mitotically cycling cells, while diffusing into the pericentromeric HP1a-marked regions in polyploid cells where centromeres are disrupted.

## The HMR C-terminus is required for HMR's physiological function in *D. melanogaster*

After verifying that the bimodal centromeric and pericentromeric localization of HMR can also be observed in the developing ovaries of *D. melanogaster* we wanted to investigate whether the C-terminus of HMR is required for HMR to fulfill its physiological function. To do this, we generated fly lines expressing full length HMR (HMR⁺) or mutant forms of it (HMR^dC, HMR²). We crossed these alleles in a mutant background (*Df(1)Hmr-*, hereafter referred to as *Hmr^ko*) and performed complementation assays to assess if the transgenic alleles are able to

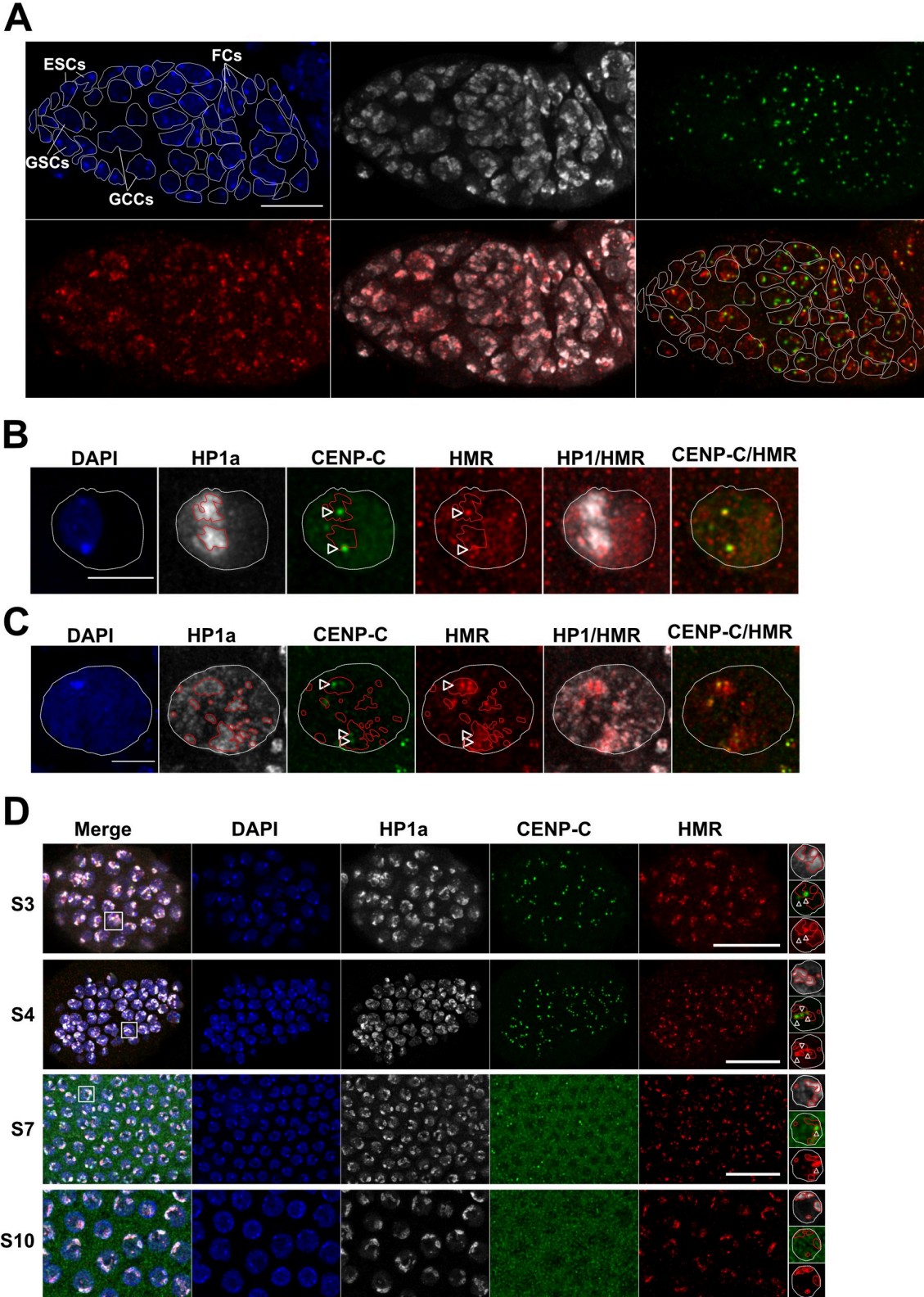

**Fig 5. HMR localization in different ovarial cell types.** Representative images of the germarium (**A**), Stage 3 oocytes (**B**), Stage 3 nurse cells (**C**) and different stages of follicle cells (**D**) from *D. melanogaster* ovarioles. Shown (from left to right) are the DAPI staining (blue) and immunofluorescent stainings using anti-HP1a (white), anti-CENP-C (green), anti-HMR (red) and the merge of either HP1a and

HMR or CENP-C and HMR channels. For better illustration the nuclear boundary is labelled with a dashed white line, the HP1a positive heterochromatic area with a red line and the centromere with a white arrow. Different cell types of the germarium were identified based on their size and the position within the germarium and labelled accordingly. Size bar indicates 10 μm in (**A**), 5 μm in (**B**) and (**C**) and 15 μm in (**D**).

rescue *Hmr* wild type functions (Fig 6A and 6B). All assays were done in ovaries, since HMR is well expressed and important for fertility and retrotransposon silencing in this tissue. After verifying the expression of the *Hmr* transgenic alleles (S6D Fig), we investigated their localization in follicle cells of sequentially developing egg chambers. In particular we looked whether the early-stage centromeric localization and the late-stage heterochromatic localization of HMR were rescued by the HMR$^{dC}$ (Figs 6C and S8). In late-stage follicle cells lacking CENP-C, truncation of HMR's C-terminus abrogates its localization to heterochromatic domains and results in a rather diffuse nuclear distribution (Fig 6C). This is in accordance with our ChIP-seq results in SL2 cells, where HMR$^{dC}$ binding to HP1a sites is substantially reduced. However, in early-stage follicle cells, unlike in SL2 cells, HMR C-terminal truncation does not affect its colocalization with centromere foci (S8 Fig). Instead, here HMR$^{dC}$ appears to be even more centromere restricted than the wild type allele.

The observation that HMR$^{dC}$ mutation disrupts the HP1a-proximal localization of HMR but not the centromeric one, supports a bimodal binding model where HMR C-terminal domain is necessary for the interaction with HP1a heterochromatin while its N-terminus mediates the interaction with centromeres. The differences between tissue culture and follicle cells might be explained by affinity differences between wild type and mutant alleles of *Hmr* that are only revealed when an endogenous copy of *Hmr* is present, like in SL2 cells. This is not the case in follicle cells, where the transgenic HMR copies are inserted in an *Hmr$^{ko}$* background and hence not in competition with endogenous HMR.

As the silencing of transposable elements (TE) has been previously shown to be impaired by *Hmr* loss of function mutations or knockdown [9,10], we tested whether the *Hmr$^{dC}$* or the *Hmr$^2$* alleles were able to restore TE silencing in an *Hmr$^{ko}$* background (Fig 6D). Whereas full length HMR was able to strongly repress all the TE studied, neither *Hmr$^{dC}$* nor *Hmr$^2$* were able to do so, showing expression levels comparable to *Hmr* deletion mutants. Since *Hmr* loss of function mutations have been shown to also cause a major reduction in female fertility [16], we also tested *Hmr$^{dC}$* for the complementation of this phenotype. Similar to what we have observed for the TE silencing, the *Hmr$^{dC}$* allele was unable to rescue the fertility defect (Fig 6E and S4 Table). All together, these results show that the *Hmr* C-terminus is required for HMR localization and physiological function in *D. melanogaster* ovaries.

## The *Hmr* C-terminus is necessary for male hybrid lethality and reproductive isolation

To understand whether the toxic *Hmr* function in interspecies hybrids also requires its C-terminus, we performed a hybrid viability suppression assay (Fig 7A). We therefore crossed *D. melanogaster* mothers carrying different *Hmr* alleles with wild type *D. simulans* stocks and counted the number of viable adult males in the offspring with respect to the total offspring (Fig 7 and S5 Table). In crosses with *Hmr* mutant mothers (*Df(1)Hmr⁻*), hybrid male offspring counts are comparable to those of females. Introduction of a wild type *Hmr* transgene (*Hmr⁺*), fully suppresses hybrid male viability, while males carrying an *Hmr$^{dC}$* or an *Hmr$^2$* allele are still viable. These results indicate that the *Hmr$^{dC}$*, similarly to *Hmr$^2$*, has lost its toxic function in hybrid males.

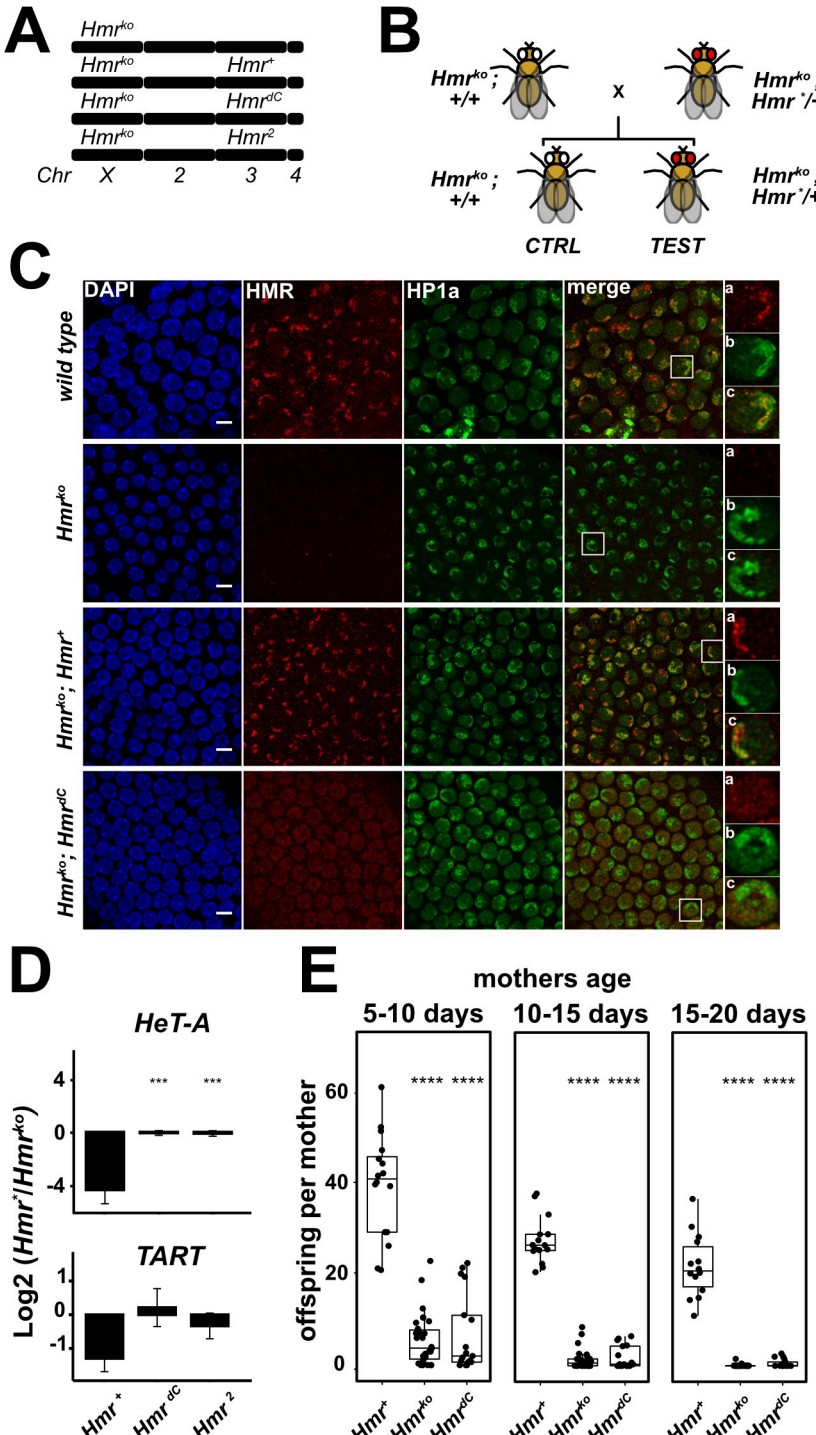

**Fig 6. The HMR C-terminus is required for HMR physiological function in *D. melanogaster*. (A)** Schematic representation of *Hmr* genotype in fly stocks used for complementation assays. *Hmr* null mutants (*Df(1)Hmr⁻*, here referred to as *Hmr^ko*) are complemented with wild type (*Hmr⁺*) or mutant (*Hmr^dC* and *Hmr²*) transgenic alleles inserted in the 3^rd chromosome. **(B)** Schematic representation of crosses performed for complementation assays. F1 siblings are compared: control animals (no *Hmr* transgene: *Df(1)Hmr⁻*; +/+) and test animals (with *Hmr* transgene: *Df(1)Hmr⁻*;*Hmr*\*/+). *Hmr*\* refers to any transgene used in this work. **(C)** Truncation of the HMR C-terminus abrogates HP1a-proximal localization of HMR *in vivo*. Representative immunofluorescence images of stage 7 follicle cells in ovarioles from *Hmr* mutant *D. melanogaster* females complemented with either of the *Hmr* transgenic alleles. Insets represent zoom into one representative cell for anti-HMR (a), anti-HP1a (b) and merge of the two channels (c). Size

bars indicate 5 μm. (**D**) Defective retrotransposons silencing in *Hmr*<sup>dC</sup> and *Hmr*<sup>2</sup>. RT-qPCR analysis measuring mRNA abundance for the telomeric retrotransposons *HeT-A* and *TART* in female ovaries. Heterozygous genotypes (*Hmr*<sup>ko</sup>; *Hmr**/+*) were compared with the respective non-complemented siblings (*Hmr*<sup>ko</sup>; +/+*). Y-axis: $\log_2$ fold-change of the mean (complemented/non-complemented) after normalization to a housekeeping gene (18s-rRNA). Error bars represent standard error of the mean (n = 3). Welch t-test was used for pairwise comparisons with *Hmr*<sup>+</sup> as a reference group and FDR for multiple testing adjustment (* $p < 0.05$, ** $p < 0.01$, *** $p < 0.001$, **** $p < 0.0001$). (**E**) HMR C-terminus is required for female fertility. Fertility defects are not complemented by *Hmr*<sup>dC</sup>: heterozygous females (*Hmr*<sup>ko</sup>; *Hmr**/+*) were compared with the respective non-complemented siblings (*Hmr*<sup>ko</sup>; +/+*). Number of adult offspring per female mother assessed in a time course (females aged 5–10, 10–15 and 15–20 days). Wilcoxon rank sum test was used for pairwise comparisons with *Hmr*<sup>+</sup> as a reference group and FDR for multiple testing adjustment (* $p < 0.05$, ** $p < 0.01$, *** $p < 0.001$, **** $p < 0.0001$). For details about fertility assays refer to S4 Table.

## Discussion

### HMR and LHR reside in a distinct six subunit complex

HMR and LHR are two well-known *Drosophila* chromatin-binding factors whose overexpression results in male lethality in hybrid animals. When expressed at native levels in *D. melanogaster*, HMR and LHR form a distinct protein complex containing 6 subunits. In addition to HMR and LHR, this complex is composed of two known histone chaperones NLP and NPH and two yet uncharacterized proteins, Buddy of HMR (BOH) 1 and 2 (CG33213 and CG4788. BOH1 is a putative transcription factor containing 4 Zn-finger DNA binding domains, which binds primarily to sites of constitutive (green) heterochromatin but also has a connection to components of (red) euchromatin [30]. Many of the BOH1 binding sites are also bound by LHR [31] and HMR [14], suggesting that BOH1 might play a role in recruiting the complex to specific genomic loci. While BOH1 contains 4 Zn-finger domains likely to bind DNA, no discernible domain can be identified in BOH2. Similar to HMR, LHR and BOH1 no ortholog of

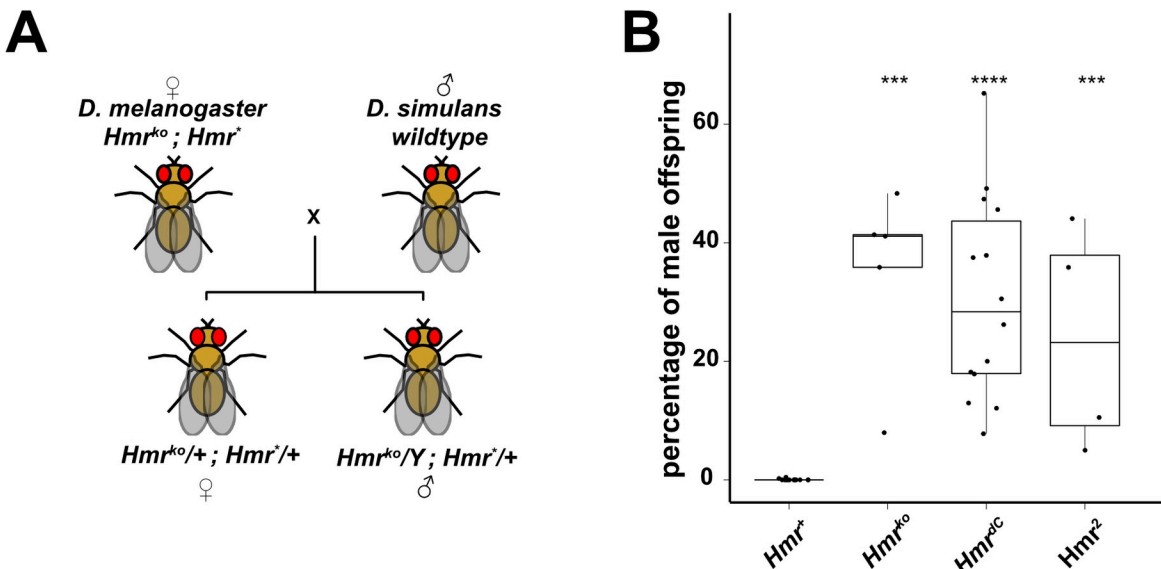

**Fig 7. Deletion of *Hmr*'s C-terminus results in failure to rescue hybrid lethality.** (**A**) Schematic representation of crosses performed for hybrid viability suppression assays. Crosses between *D. melanogaster* mothers carrying *Hmr* transgenes in a *Df(1)Hmr*—background (*Hmr*<sup>ko</sup>; *Hmr**) and wild type *D. simulans* males (C167.4 and *w*<sup>501</sup>). Suppression of hybrid male viability is analyzed within the transgene-carrying offspring only. *Hmr** refers to any transgene used in this work. (**B**) HMR C-terminus is necessary for HMR lethal function in hybrid males. Suppression of hybrid males viability was measured as percentage of viable adult males in the total hybrid adult offspring. Crosses from non-complemented *Hmr*<sup>ko</sup> mothers were used as control. Dots represent individual biological replicates. Wilcoxon rank sum test was used for pairwise comparisons with *Hmr*<sup>+</sup> as a reference group and fdr for multiple testing adjustment (* $p < 0.05$, ** $p < 0.01$, *** $p < 0.001$, **** $p < 0.0001$). For details refer to S5 Table.

BOH2 can be identified outside the dipteran lineage and no molecular analysis of it has been done so far. Here, we show that BOH2 interacts with HMR and LHR under native conditions but also with a set of other nuclear factors. *Nlp* and *Nph* are the two *Drosophila* paralogues of the nucleoplasmin family of histone chaperones, which are important for sperm decondensation upon fertilization [32], chromosome pairing and centromere clustering [33,34]. Both proteins depend on HMR to localize to the border between centromeric and pericentromeric chromatin [15].

## Excess HMR and LHR interact with novel chromatin factors

As hybrid animals suffer from increased levels of HMR and LHR, we performed AP-MS experiments of ectopically expressed HMR/LHR in SL2 cells in the presence of endogenous levels of HMR/LHR. This strategy allowed us to preferentially isolate proteins that interact with surplus molecules of HMR and LHR. Indeed, ectopically expressed HMR and LHR bind to several heterochromatic factors which are not detected under native conditions. Among many Zn-finger containing proteins that may explain the disperse localization of overexpressed HMR on polytene chromosomes [9] we observe a stable interaction of the extra HMR/LHR molecules with GFZF, another factor required for male hybrid lethality [8]. This finding is consistent with HMR and GFZF aberrantly colocalizing in interspecies hybrids and upon HMR/LHR overexpression [11].

## HMR contains two functionally important protein-protein interaction modules

Our proteomic analysis of *Hmr* mutants suggests that HMR's N-terminal MADF3 domain, which is mutated in the *Hmr²* allele, mediates the interaction with NLP, NPH, BOH1 and BOH2 while its C-terminus binds LHR, and through this interaction presumably recruits HP1a [9,16,24,35]. Further genome-wide and cytological experiments with these *Hmr* alleles reveal that the integrity of these interactions as well as a balanced expression of *Hmr/Lhr* is vital for HMR proper targeting, its physiological function, and its ability to kill hybrid males. In particular both our ChIP Seq data and ovaries stainings in early follicle cells show that HMR's C-terminus is necessary for localization to heterochromatin. Additionally, ovary stainings in follicle cells up to stage S7, where HMR wild type localization is centromeric, show that HMR'S C-terminus is instead dispensable for it's binding to the centromere.

The latter finding is in apparent contradiction with the stainings we performed in SL2 cells, where HMR$^{dC}$ mutation disrupts centromeric localization. However, it is worth noticing that the experiments in SL2 cells were done in the presence of endogenous (wild type) HMR, and therefore reflect a competitive situation between HMR alleles, which could result in a more sensitive readout, revealing subtle differences between wild type and mutant HMR. The loss of HMR$^{dC}$ binding to centromeres in SL2 cells, is presumably happening because CENP-C, which is required for HMR's recruitment at centromeres [12], is already saturated by the endogenous HMR. In contrast, we found a very centromere restricted localization of HMR in mitotically cycling follicle cells, in which the transgenic *Hmr* allele was the sole source of HMR.

These findings support a model in which HMR is recruited to centromeres by CENP-C in mitotically cycling cells. However, in absence of this recruitment HMR's C-terminus interacts with HP1-a containing heterochromatin, resulting in a pericentromeric localization instead.

The functional assays in flies showed that both mutants (*Hmr²* and *Hmr$^{dC}$*) are not able to rescue the *Hmr$^{ko}$* phenotype. These findings support the hypothesis that HMR needs to interact with heterochromatic factors as well as components of the chromocenter to accomplish its

function. We therefore consider it unlikely that the *Hmr* mutant phenotypes observed in *D. mel* (reduced female fertility, upregulation of TEs) are solely dependent on HMR's ability to bind heterochromatin, since the mutant HMR$^2$ protein is still able to interact with LHR and HP1a but still does not rescue these phenotypes. At the same time, HMR's localization to the chromocenter is not sufficient to achieve full functionality, as the HMR$^{dC}$ protein still localizes to the chromocenter in absence of a wild type copy but nevertheless fails to rescue the fertility defect. We therefore propose that HMR organizes the chromocenter by directly interacting with centromeric as well as heterochromatic factors and that both interactions are required for HMR to fulfil its function. In fact, defects in chromocenter bundling have been shown to result in micronuclei formation and loss of cellular viability in the imaginal discs and lymph glands [36,37], a phenotype that is also observed in interspecies hybrids [19,20].

## HMR's dual binding may be required for hybrid male lethality

Mutations impairing HMR's ability to bind either heterochromatin or the centromere not only fail to complement *Hmr* null phenotypes in *D. mel*, but also no longer cause lethality of male hybrids of *D.mel* and *D.sim*. However, in contrast to the phenotypes discussed above, hybrid lethality is a consequence of overexpression rather than a loss of HMR and LHR [9]. Due to the technical difficulty to study the HMR complex in prematurely dying hybrid male flies, we simulated the hybrid situation by overexpressing HMR and LHR in *D.mel* tissue culture cells. As we had shown in the past that HMR$_{mel}$ and LHR$_{mel}$ interact with a very similar set of proteins to their *D.sim* counterparts when expressed in SL2 cells [9], we think their overexpression in this cell system constitutes a useful proxy for the hybrid situation. A comparison of the native HMR interaction proteome with the one of overexpressed HMR, revealed that overexpression leads to novel interactions with known chromatin factors such as BEAF-32, CP190, PROD, HIPP1 or GFZF [9,11], which has been shown to be important for hybrid male lethality [8].

These newly gained protein-protein interactions we observe in the presence of excessive HMR and LHR are probably the reason for an aberrant targeting of the complex, leading to a possible mislinkage of genomic loci and a failure to properly regulate the chromocenter in mitotically dividing cells. In combination, these effects will eventually result in defects in cell cycle progression [13,20]. Consistently, as neither of the *Hmr* mutations described here is able to simultaneously bind both heterochromatin and centromeric chromatin components, the expression of these alleles does not result in hybrid lethality. Interestingly, Jagannathan and Yamashita have recently shown [38] that hybrid incompatibility factors HMR and LHR lead to chromocenter disruption in hybrids, suggesting a complex interplay between the HMR complex and other factors that must be finely tuned to maintain functional chromocenter and viable cells, a condition that is not met in hybrids.

While we still do not understand the detailed molecular mechanism that mediates the physiological function of the identified HMR complex, our results suggest that its ability to interact with two types of chromatin is of critical importance. It may very well be that the increased levels of HMR and LHR in hybrids pick up novel additional chromatin proteins thereby unleashing potentially lethal chromatin driver systems that evolved differently in the two closely related species.

The isolation of a defined complex involving the hybrid incompatibility proteins HMR and LHR will allow a more detailed molecular analysis of its function and sets the ground for future comparative studies on the divergent evolution of its components within species and on their lethal interactions in hybrids.

## Supporting information

**S1 Fig. Interactomes of HMR/LHR interactors provide evidence for the existence of a defined HMR protein complex (related to Fig 1). (A)**—**(D)** HMR complex components are consistently enriched in reciprocal IPs. Volcano plots showing the interactome resulting from the AP-MS of the HMR complex components BOH1 (n = 4), BOH2 (n = 5), NLP (n = 3) and HP1a (n = 4) against a "mock" purification. The HMR complex subunits are labeled in red. X-axis: $\log_2$ fold-change of factor enrichment in IPs against mock purification (CTRL). Y-axis: significance of enrichment given as–$\log_{10}$ p-value calculated with a linear model. A list of the unlabeled additional bait-specific interactors is provided in S2 Table. **(E)** Network plot showing a highly connected HMR complex surrounded by subunit-specific interactors. Enriched proteins from each AP-MS experiment from HMR complex components were first selected (cut off: log2FC > 2.5, p-adjusted < 0.05) and integrated in an interaction network drawn with force directed layout in D3.js and R. Nodes represent proteins significantly interacting with at least one of the HMR complex components. Edges represent physical connections experimentally detected in this work. HMR complex subunits (and baits) are labelled in red. Interactive volcano plots and interaction network are available at the following (URL).
(TIFF)

**S2 Fig. The HMR complex components BOH1 and BOH2 largely colocalize with HMR in proximity to centromeres and HP1a domains (related to Fig 1). (A)** BOH1 and BOH2 colocalize with HMR and CID in SL2 cells. Immunofluorescence images of cells expressing FLAG-HA-BOH2 (upper panel) or FLAG-HA-BOH1 (lower panel) showing the co-staining of HA-BOH2 or HA-BOH1, respectively, with CENP-A (centromeres) and HMR. **(B)** BOH1 and BOH2 colocalize with HP1a and CID in SL2 cells. Immunofluorescence images of cells expressing FLAG-HA-BOH2 (upper panel) or FLAG-HA-BOH1 (lower panel) showing the co-staining of HA-BOH2 or HA-BOH1, respectively, with CID (centromeres) and HP1a (pericentromeric chromatin). For **(A)** and **(B)** size bar indicates 3 μm, DAPI staining indicates nuclei.
(TIFF)

**S3 Fig. Excess of HMR interacts beyond the HMR core complex with chromatin architecture proteins (related to Fig 2). (A)** Volcano plot highlighting novel interactions gained by HMR upon overexpression. X-axis: $\log_2$ fold-change of FLAG-HMR$^{endo}$ IPs (right side of the plot) vs FLAG-HMR$^+$ IPs (left side of the plot). Y-axis: significance of enrichment given as–$\log_{10}$ p-value calculated with a linear model. HMR core complex subunits are labelled in red, novel factors enriched upon HMR overexpression are labelled in blue. Unlabeled additional bait-specific interactors are listed in S3 Table. **(B)** GO terms enriched upon overexpression of HMR. In **(A)** and **(B)** proteins were labelled or considered for GO search only if enriched in HMR$^+$ or HMR$^{endo}$ vs CTRL (p < 0.05) and differentially enriched between HMR$^+$ and HMR$^{endo}$ ($\log_2$ fold-change (HMR$^{endo}$/HMR+) < 1.5).
(TIFF)

**S4 Fig. Two different Hmr mutations interfere differently with HMR interactome and HMR complex formation (related to Fig 3). (A)** Recombinantly co-expressed HMR$^{dC}$ and LHR do not interact. Western blot showing anti-HA immunoprecipitation in nuclear extracts from Sf21 insect cells transfected with HA-HMR and His-LHR. IP performed with anti-HA antibody and western blot probed with anti-HMR or anti-LHR antibodies. **(B)** HMR C-terminus is required for HMR interaction with LHR and HP1a in *D. mel* SL2 cells. Western blot showing HMR immunoprecipitation in SL2 cells stably transfected with either full length HMR or a C-terminally truncated HMR$^{dC}$ along with Myc-LHR. IP performed with anti-

FLAG antibody targeting FLAG-HMR and western blot probed with anti-HA (HMR), anti-Myc (LHR) and anti-HP1a. **(C)** Western blot showing LHR immunoprecipitation in SL2 cells stably transfected with either full length HMR or a C-terminally truncated HMR$^{dC}$ along with Myc-LHR. IP performed with anti-Myc antibody targeting Myc-LHR and western blot probed with anti-FLAG (HMR) and anti-LHR. **(D)** Volcano plot highlighting interactions depleted in HMR$^{dC}$. X-axis: log$_2$ fold-change of FLAG-HMR$^{dC}$ IPs (right side of the plot) vs FLAG-HMR$^+$ IPs (left side of the plot). Y-axis: significance of enrichment given as–log$_{10}$ p-value calculated with a linear model. HMR complex subunits are labelled in red. In blue are factors depleted upon HMR$^{dC}$ mutation (among the endogenous or overexpression-induced interactions of HMR). Unlabeled additional bait-specific interactors are listed in S3 Table. **(E)** GO terms depleted upon HMR$^{dC}$ mutation. **(F)** Volcano plot highlighting interactions depleted in HMR$^2$. X-axis: log$_2$ fold-change of FLAG- HMR$^2$ IPs (right side of the plot) vs FLAG- HMR$^+$ IPs (left side of the plot). Y-axis: significance of enrichment given as–log$_{10}$ p-value calculated with a linear model. HMR complex subunits are labelled in red. In blue are factors depleted upon HMR$^2$ mutation (among the endogenous or overexpression-induced interactions of HMR). Unlabeled additional bait-specific interactors are listed in S3 Table. **(G)** GO terms depleted upon HMR$^2$ mutation. In **(D)** and **(G)** proteins labelled or used for GO search include endogenous or overexpression-induced interactions of HMR (i.e. enriched in HMR$^+$ or HMR vs CTRL with p < 0.05) and differentially enriched between HMR$^+$ and the HMR mutant analyzed (log$_2$ fold-change (HMR*/HMR+) < 1.5).
(TIFF)

**S5 Fig. The HMR C-terminus is required for HMR localization in proximity to centromeres and HP1a-bound chromatin (related to Fig 4). (A)** Heatmaps of ChIP-seq profiles (z-score normalized) centred at high confidence FLAG-HMR peaks in 4 kb windows. Peaks are grouped by HP1a class and sorted by the ChIP signal in native HMR ChIP. From left to right, anti-HMR ChIP in untransfected cells, anti-HMR ChIP in cells transfected with FLAG-*Hmr*$^+$ and FLAG-*Hmr*$^{dC}$, anti-FLAG ChIP of cells transfected with FLAG-*Hmr*$^+$ or FLAG-*Hmr*$^{dC}$, and anti-HP1a and anti-CP190. The latter two are representative of the two classes of HMR peaks: HP1a-proximal and non-HP1a-proximal. **(B)** Chromosome-wide FLAG-HMR ChIP-seq profiles (z-score normalized) for *Hmr*$^+$ (light blue), *Hmr*$^{dC}$ (dark blue) and HP1a (green). Chromosomes X and 4 are shown. **(C)** *Hmr*$^{dC}$ is depleted at heterochromatin rich chromosome 4. Percentage of FLAG-HMR ChIP-seq peaks located on chromosome 4 for each replicate (n = 5). *Hmr*$^+$ (light blue) and *Hmr*$^{dC}$ (dark blue) are shown. P-values are obtained by a linear model. FLAG-HMR plots represent an average of 5 biological replicates.
(TIFF)

**S6 Fig. The HMR C-terminus is required for HMR localization in proximity to centromeres and HP1a-bound chromatin (related to Fig 4A and 5). (A)** The HMR C-terminus is required for HMR to form bright centromeric foci. Quantification of the percentage of centromeric foci (marked by CENP-C) associated with HMR in imunofluorescent stainings in SL2 cells expressing different *Hmr* transgenes (*Hmr*$^+$, *Hmr*$^{dC}$ and *Hmr*$^2$). Stainings were performed with DAPI, anti-HA (recognizing HA-HMR) and anti-CENP-C antibodies. For staining details refer to Fig 4A. **(B)** The number of centromeric foci per cell inversely correlates with HMR's association with centromeres. Scatter plot displaying the relation between the percentage of centromeric foci associated with HMR (x-axis, binned by 10% units) vs number of centromeric foci per cell (y-axis). Each dot represents a measured cell (a pool of all experiments from all Hmr alleles is displayed). **(C)** The ectopic expression of *Hmr* mutants correlates with higher numbers of centromeric foci. Boxplots displaying the number of centomeric foci per cell (y-axis) for each of the Hmr alleles (x-axis). Each dot represents a measured cell (a

pool of all stained cells for each allele). (**D**) Protein expression in ovaries from *Hmr^ko^* stocks complemented with different *Hmr* transgenes. Western blot probed with anti-HMR, anti-LHR and anti-LAMIN antibodies.
(TIFF)

**S7 Fig. HMR's distribution varies dependent on the cell type (related to Fig 5A). (A)** Identical figure as 5A but with enlarged cells numbered. Shown are the DAPI staining (blue) and immunofluorescent stainings using anti-HP1a (white), anti-CENP-C (green), anti-HMR (red) and the merge of all channels. (**B**) For better illustration of the distribution of the different HMR related proteins, single cells were enlarged. In these insets the nuclear boundary is labelled with a dashed white line, the HP1a positive heterochromatic area with a red line and the centromere with a white arrowhead. Size bar indicates 10 μm in (**A**) and 3 μm in (**B**).
(TIFF)

**S8 Fig. Localization of different HMR alleles in early follicle cells (related to Fig 6C).** Representative images of different HMR alleles in stage 4 follicle cells. Shown are the DAPI staining (blue) and immunofluorescent stainings using anti-HP1a (white), anti-CENP-C (green), anti-HMR (red) and the merge of all channels (leftmost panels). For better illustration of the distribution of the HMR variants, a single follicle cell was enlarged and depicted on the rightmost panels. In these insets the nuclear boundary is labelled with a dashed white line, the HP1a positive heterochromatic area with a red line and the centromere with a white arrowhead. Size bar indicates 15 μm.
(TIFF)

**S1 Table. Summary of antibodies, primers, cell lines and fly stocks used in this study.**
(XLSX)

**S2 Table. Enrichment values of proteins of AP-MS experiments using all subunits of the HMR complex (related to Fig 1).**
(XLSX)

**S3 Table. Enrichment values of proteins of HMR AP-MS experiments upon overexpression (related to Fig 2 and 3).**
(XLSX)

**S4 Table. Data of fertility assays (related to Fig 6).**
(XLSX)

**S5 Table. Data of hybrid crossings (related to Fig 7).**
(XLSX)

**S1 Methods. Extended description of supplementary methods.**
(DOCX)

## Acknowledgments

We thank Patrick Heun, Christian Lehner and Harald Saumweber for the Nlp, CENP-A, CENP-C and Lamin antibodies respectively. We also thank Daniel Barbash for the Df(1)Hmr- fly strain. We furthermore would like to thank Catherine Regnard, Sandro Baldi and Alessandro Scacchetti for critical comments on the manuscript. Andreas Schmidt and Marc Wirth for advice on proteomic analysis and for mass-spectrometry analysis. Irene Vetter and Silke Krause for cloning of expression constructs. Thomas Gerland, Raffaella Villa and Alessandro Scacchetti for advice on ChIP-seq, Angelika Zabel on ChIP-seq library preparation and

Kenneth Boerner for advice on ovary staining. We also would like to thank Sophia Groh for graphical aid and Elisabeth Schröder-Reiter and Markus Hohle for constant support. We also would like to thank the entire Imhof group, Peter Becker and the Becker department for helpful discussions. In addition, we would also like to thank Stefan Krebs and Helmut Blum from LAFUGA facility for sequencing.

## Author Contributions

**Conceptualization:** Andrea Lukacs, Andreas W. Thomae, Axel Imhof.

**Data curation:** Andrea Lukacs, Andreas W. Thomae, Tamas Schauer.

**Formal analysis:** Andreas W. Thomae.

**Funding acquisition:** Axel Imhof.

**Investigation:** Andrea Lukacs, Andreas W. Thomae, Peter Krueger, Tamas Schauer, Anuroop V. Venkatasubramani, Natalia Y. Kochanova, Rupam Choudhury, Ignasi Forne.

**Methodology:** Andrea Lukacs, Andreas W. Thomae, Peter Krueger, Tamas Schauer, Anuroop V. Venkatasubramani, Natalia Y. Kochanova, Rupam Choudhury, Ignasi Forne.

**Project administration:** Andrea Lukacs, Axel Imhof.

**Resources:** Andrea Lukacs, Axel Imhof.

**Software:** Tamas Schauer, Anuroop V. Venkatasubramani, Wasim Aftab.

**Supervision:** Andrea Lukacs, Andreas W. Thomae.

**Validation:** Andrea Lukacs, Anuroop V. Venkatasubramani.

**Visualization:** Andrea Lukacs, Andreas W. Thomae, Tamas Schauer, Anuroop V. Venkatasubramani, Natalia Y. Kochanova, Wasim Aftab, Ignasi Forne, Axel Imhof.

**Writing – original draft:** Andrea Lukacs, Axel Imhof.

**Writing – review & editing:** Andrea Lukacs, Andreas W. Thomae, Tamas Schauer, Anuroop V. Venkatasubramani, Natalia Y. Kochanova, Wasim Aftab, Ignasi Forne, Axel Imhof.

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
