## [Editor Report · Decision Letter 0]

15 Jun 2021

Dear Dr Imhof,

Thank you very much for submitting your Research Article entitled 'The Integrity of the HMR complex is necessary for centromeric binding and reproductive isolation in Drosophila' to PLOS Genetics.

The manuscript was fully evaluated by independent peer reviewers at Review Commons. The reviewers appreciated the attention to an important problem, but raised some substantial concerns about the current manuscript. Based on your proposed responses to the reviewer comments, we would be willing to review a much-revised version. We cannot, of course, promise publication at that time as the number of changes will require a re-review from at least one of the original reviewers.

If you decide to revise the manuscript for further consideration at PLOS Genetics, please aim to resubmit within the next 60 days, unless it will take extra time to address the concerns of the reviewers, in which case we would appreciate an expected resubmission date by email to plosgenetics@plos.org.

[LINK]

We are sorry that we cannot be more positive about your manuscript at this stage. Please do not hesitate to contact us if you have any concerns or questions.

Yours sincerely,

Harmit S. Malik

Associate Editor

PLOS Genetics

Kirsten Bomblies

Section Editor: Evolution

PLOS Genetics

---

## [Decision Letter · Decision Letter 1]

21 Jul 2021

Dear Dr Imhof,

Thank you very much for submitting your Research Article entitled 'The Integrity of the HMR complex is necessary for centromeric binding and reproductive isolation in Drosophila' to PLOS Genetics.

The manuscript was fully evaluated at the editorial level and by independent peer reviewers who previously evaluated your paper. All reviewers are now largely satisfied but each identified some concerns that we ask you address in a revised manuscript- in particular you to be clear when referring to 'hybrid' phenotypes that are inferred indirectly and to make the distinction between transgenes and endogenously expressed genes.

We therefore ask you to modify the manuscript according to the review recommendations. Your revisions should address the specific points made by each reviewer.

[LINK]

Yours sincerely,

Harmit S. Malik

Associate Editor

PLOS Genetics

Kirsten Bomblies

Section Editor: Evolution

PLOS Genetics

Reviewer's Responses to Questions

**Comments to the Authors:**

Reviewer #1: The manuscript “The integrity of the speciation core complex is necessary for centromeric binding and reproductive isolation in Drosophila” by Lukacs and colleagues was previously reviewed via Review Commons by these reviewers. Therefore, we focused on how the concerns raised were addressed, whether the manuscript has improved and whether any outstanding issues still remain that would preclude publication in PLoS Genetics.

In our view, the concerns raised by all three reviewers have been adequately addressed.

The following specific points have been addressed to our satisfaction:

1) The overall flow of the manuscript reads much better and is less speculative.

2) The addition of the IF study on how HMR localizes with CENP-C or HP1a in ovaries are an important addition, which help clarify the dynamics of HMR binding in development and explain how these dynamics correlate with TE suppression.

3)The finding that the CTD of HMR is important for HMR localization to pericentromeric HP1a provides an important mechanistic insight.

4) The observation that HRM binds two distinct chromatin domains, both of which are necessary for proper development, are insightful from the dynamic perspective of chromatin-binding factors.

Overall, these data imply that both spatiotemporal regulation of chromatin factors and interaction with other chromatin-associated factors are critical for establishing and maintaining chromatin function and subsequent downstream effects. It will be interesting to learn how future studies will tease apart how the binding of HMR to these two chromatin domains are regulated. The updated discussion succinctly captures this point.

A few minor points could benefit from editing:

+ On page 16, the sentence “To have a more comprehensive of HMR’s localization in flies …" does not flow well.

+ On page 17, the use of the word “flies” in “… can also be observed in flies, we wanted …” is a little broad. Maybe write out more specifically which fly they are referring to?

Reviewer #2: Review is uploaded as an attachment.

Reviewer #3: I’ve looked through previous reviews and responses to them. It is somewhat unclear exactly at what stage I was brought in to review this at PLoS Genetics.

In previous version(s?) of this manuscript, the reviewers generally agreed that the data quality is good but the interpretations are not well-supported. In this revised version, the authors have improved the writing considerably, and the overall conclusions are reasonably supported. Specific comments below cover several additional issues along this line, but once they are addressed, this manuscript will be of considerable interest for the field.

-Page 13-14: They infer that the pulldown experiment under Hmr overexpression condition mimics hybrid condition. However, it should be noted that this is still in pure species condition (absence of any simulans proteins). Of course, they don’t need to do such experiments, but this difference needs to be clearly mentioned.

-Figure2, referring Hmr ectopic expression to as ‘Hmr+’ is very misleading. I understand this means ‘wild type transgene’, but in the context of differentiating Hmr-endo vs. Hmr+, it’s hard to understand ‘Hmr+’ indicates ectopic expression.

-Please label Figure 5A.

-Fig7 title ‘Hmr C-ter is required for hybrid lethality’ is misleading. Although this statement is not wrong, the results might simply indicate that Hmr-deltaC is just non-functional, as is shown in Figure 6. But the figure title/subheading indicate a little more than that (as if it is specifically required for hybrid lethality, without affecting its endogenous function in pure species context). I recommend a slightly more cautious description.

**Have all data underlying the figures and results presented in the manuscript been provided?**

Reviewer #1: Yes

Reviewer #2: Yes

Reviewer #3: Yes

PLOS authors have the option to publish the peer review history of their article (what does this mean?). If published, this will include your full peer review and any attached files.

Reviewer #1: **Yes: **Yamini Dalal and Daniel Melters

Reviewer #2: No

Reviewer #3: **Yes: **Yukiko Yamashita

---

## [Editor Report · Decision Letter 2]

27 Jul 2021

Dear Dr Imhof,

We are pleased to inform you that your manuscript entitled "The Integrity of the HMR complex is necessary for centromeric binding and reproductive isolation in Drosophila" has been editorially accepted for publication in PLOS Genetics. Congratulations!

Yours sincerely,

Harmit S. Malik

Associate Editor

PLOS Genetics

Kirsten Bomblies

Section Editor: Evolution

PLOS Genetics

Comments from the reviewers (if applicable):

**Data Deposition**

http://datadryad.org/submit?journalID=pgenetics&manu=PGENETICS-D-21-00788R2

**Press Queries**

---

## [Editor Report · Acceptance letter]

18 Aug 2021

PGENETICS-D-21-00788R2 

The Integrity of the HMR complex is necessary for centromeric binding and reproductive isolation in Drosophila 

Dear Dr Imhof, 

We are pleased to inform you that your manuscript entitled "The Integrity of the HMR complex is necessary for centromeric binding and reproductive isolation in Drosophila" has been formally accepted for publication in PLOS Genetics! Your manuscript is now with our production department and you will be notified of the publication date in due course.

With kind regards,

Livia Horvath

PLOS Genetics

On behalf of:
